# Qualitative study to explore UK medical students' and junior doctors' experiences of occupational stress and mental health during the COVID-19 pandemic

James Tawse  ,[1,2] Evangelia Demou  [3]

¹University of Glasgow College of Social Sciences, Glasgow, UK
²Staff Advice and Liaison Service, Alder Hey Children's Hospital, Liverpool, UK
³MRC/CSO Social and Public Health Sciences Unit, University of Glasgow College of Medical Veterinary and Life Sciences, Glasgow, UK

**Correspondence to**
James Tawse;
james.tawse@alderhey.nhs.uk

## ABSTRACT

**Objectives** This qualitative study aimed to explore the occupational experiences of medical students and junior doctors working during the COVID-19 pandemic. In particular, the research sought to identify factors which mediated work stress, barriers to disclosing mental health problems and levels of support medical students and junior doctors received during the pandemic.

**Design** This study was a form of thematic analysis and adopted an inductive, 'bottom-up' approach, in which coded categories were derived from rich, descriptive data.

**Setting** Semistructured interviews were conducted online with UK-based medical students and junior doctors. Interviews were recorded, and analysis was done by coding salient quotes into themes.

**Participants** The final sample consisted of seven junior doctors and eight medical students, during the summer of 2021.

**Results** High levels of occupational stress were identified, which were exacerbated by COVID-19. A number of organisational difficulties associated with the pandemic compounded participants' experiences of work stress. Participants recognised progress towards promoting and managing mental health within the profession but may still be reluctant to access support services. Barriers to disclosure included fear of stigmatisation, concerns about adding to colleagues' workloads, lack of clarity about career implications and mistrust of occupational health services.

**Conclusions** While attitudes towards mental health have improved, medical students and junior doctors may avoid seeking help. Given the immense pressures faced by health services, it is imperative that extra measures are implemented to minimise work-stress, encourage help-seeking behaviours and promote supportive work cultures.

## INTRODUCTION

COVID-19 is one of the most severe international health problems in the last decades.[1] Communities worldwide suffered directly from high infection rates associated with increased mortality, and indirectly through measures to contain the spreading of the virus.[2] In the UK, education, workplaces and health systems had to radically adjust their working practices to adhere to social distancing, safety and health guidelines.[3 4] For medical students and junior doctors, the pandemic presented an unexpected, disruptive external shock which created uncertainty and distress.[5 6]

Since the onset of the pandemic, medical students experienced drastic changes to their academic curricula.[7] Most medical schools shifted to online learning with a reduction or even suspension of practical teaching.[5] Medical students were faced with the discontinuation of clinical placements, novel examination formats and studying in isolation.[8] In response to the public health crisis, medical students across the UK volunteered in hospitals and general practices, to support shortages in medical staff.[7] However, this placed volunteers at an increased risk of infection, and presented challenges to provide students with adequate supervision.[9] Furthermore, UK governing bodies allowed final year medical students to graduate 6–8

### STRENGTHS AND LIMITATIONS OF THIS STUDY

⇒ This research captured medical students' and junior doctors' experiences of working during the summer of 2021, alongside rising rates of COVID-19 in the UK, ongoing restrictions to social contact and unprecedented pressure on health services.
⇒ This study accessed a unique cohort of medical students and junior doctors, who did not receive standard training opportunities.
⇒ Thematic analysis was conducted on the interview data, facilitating the development of comprehensive and insightful themes.
⇒ The representativeness and transferability of the findings may be limited, as most participants were males and of a white British ethnicity.

weeks early from medical school to join the frontline in fighting against the pandemic.[10] This raises questions about this group of newly graduated doctors' preparedness to enter the profession, under difficult circumstances, and is concerning given that less prepared trainees make more medical errors and suffer from higher rates of burn-out.[11]

Junior doctors working to fight COVID-19 reported stress and distress due to the strain on healthcare systems.[6] Such stressors included understaffing, increased workload and uncertainty due to major changes to working practices.[12 13] On top of this, healthcare professionals' resilience was hindered by loss of support, infection of friends and family, as well as exposure to unprecedented levels of suffering and illness, both of colleagues and patients.[14]

Before the pandemic, concerns had already been raised about medical students' and junior doctors' well-being.[15–18] Research had consistently shown that medical students and junior doctors experience significant levels of work-stress and burnout, as well as high rates of mental health problems and suicide when compared with the general population.[18–21]

In recent times, systemic issues within healthcare have compounded workers' mental health difficulties.[22–24] Poor working conditions, inadequate support mechanisms and a lack of flexibility have contributed towards record numbers of trainee doctors leaving the profession altogether.[25]

The mental health burden of junior doctors and medical students working during the pandemic has been the subject of interest in quantitative[12 13 26–28] and qualitative research studies.[6 13 29–34] However, to the author's knowledge, no qualitative studies have simultaneously examined the mental health of medical students and junior doctors during COVID-19. As such, this paper can offer valuable insights into the relative stress, mental well-being and access to support of these groups, during this difficult transitionary period from student to doctor,[35] and in unprecedented circumstances. This qualitative study captures their voices at a distinct and testing time, among rising rates of COVID-19 in the summer of 2021 and the long-term impact of COVID-19 on healthcare workers' well-being remains unclear.[36 37] Consequently, continued efforts to protect medical students' and junior doctors' well-being are warranted. In doing so, retention issues may be addressed, which threaten patient safety, and the future of health services.[25]

This study aimed to explore medical students' and junior doctors' experiences of occupational stress and mental health during the COVID-19 pandemic in the UK. Specifically, this study addressed the following three research questions. The research questions informed the study and the interview guide (online supplemental appendix 1), they were not directly asked to the participants.

**RQ1**: What are the key sources of occupational stress among medical students and junior doctors working during the COVID-19 pandemic?

**RQ2**: What barriers exist to accessing mental health support within this workforce?

**RQ3**: Have medical students and junior doctors received sufficient support to manage their work stress and mental health challenges during the COVID-19 pandemic?

## METHODS

### Participant recruitment

The inclusion criteria for the study were medical students and junior doctors working in the UK, during the summer of 2021. For this study, 'junior doctor' refers to a doctor who graduated from medical school in the last 5 years, and is currently enrolled on a medical training programme. Medical students were only invited to take part if they were in fourth year onwards. This cut-off was selected as these students had experience of clinical placement, and they faced significant disruption to their training, in close proximity to qualification. A combination of sampling strategies were used to recruit participants.[38] Initially, purposeful sampling of participants was implemented, where the lead researcher reached out to personal contacts and acquaintances who may have been available and willing to participate. Following this, participants were recruited using snowball sampling.[39 40] We aimed to conduct approximately 15–20 interviews.

Once sampled, participants were contacted to arrange one-to-one interviews and interviews were arranged for a time that suited the participants.[6] Before starting the interview, participants were made aware of the requirements and purposes of the research, the identity of the researcher and how the results would be used.[41 42] Participants were reminded of their ethical rights surrounding confidentiality, data protection and their right to withdraw at any point.[43] A safety plan was in place, to ensure that support could be provided in the event of a concerning disclosure.[6 44] Following this, participants made an informed decision on their participation.

Semistructured, one-on-one interviews were conducted online. The research questions were used as guides to develop the interview schedule (online supplemental appendix 1). Interviews took place over Zoom, to adhere to the social distancing guidelines in place (July–August 2021).[45] Recordings of interviews were taken using the built-in tool on Zoom, for subsequent verbatim transcription. After transcription, the recordings were erased. Interviews lasted an average of 36 min (range: 25–52 min).

The Standards for Reporting Qualitative Research reporting guidelines were used (online supplemental appendix 2).[46]

### Patient and public involvement

Patients or the public were not involved in the design and conduct of this study.

### Data analysis methodology

An inductively driven thematic analysis of the interview data was undertaken, in accordance with Braun and

**Table 1** Sample characteristics

| Participant ID | Training level | Place of work/study | Gender identity | Ethnicity |
|---|---|---|---|---|
| #1 | 5th year medical student | North-West England | Male | White British |
| #2 | 5th year medical student | North-West England | Male | White British |
| #3 | 4th year medical student | North-East England | Male | White British |
| #4 | Foundation Year 1 (FY1) | Scotland | Male | White British |
| #5 | FY1 | Midlands England | Male | White British |
| #6 | FY1 | Midlands England | Female | White British |
| #7 | 5th year medical student | North-West England | Male | White British |
| #8 | 4th year medical student | South-East England | Male | White British |
| #9 | FY1 | Scotland | Male | White British |
| #10 | FY3 | North-East England | Female | White British |
| #11 | FY1 | North-West England | Male | Asian British |
| #12 | 5th year medical student | North-West England | Male | White British |
| #13 | 5th year medical student | Scotland | Male | White British |
| #14 | 4th year medical student | Scotland | Male | White British |
| #15 | FY3 | South-East England | Female | Asian British |

Clarke's six-stage model.[47] The dependability of the findings was assured by conducting regular meetings with the second researcher (ED) to produce a coding framework that accurately captured the emerging themes. Data analysis took place in parallel with interviews, which meant that the researcher could closely follow emerging themes and confirm when no new themes were arising.[48] NVivo V.12 was used to analyse the data.

## RESULTS

Participant demographic information and the thematic map are presented in table 1 and figure 1, respectively. The final sample consisted of 15 participants (8 medical students, 7 junior doctors). Of the medical students, three were in their fourth year of study, while five were in their fifth year of study. With regards to the junior doctors, five were in their first year of postgraduate work, referred to

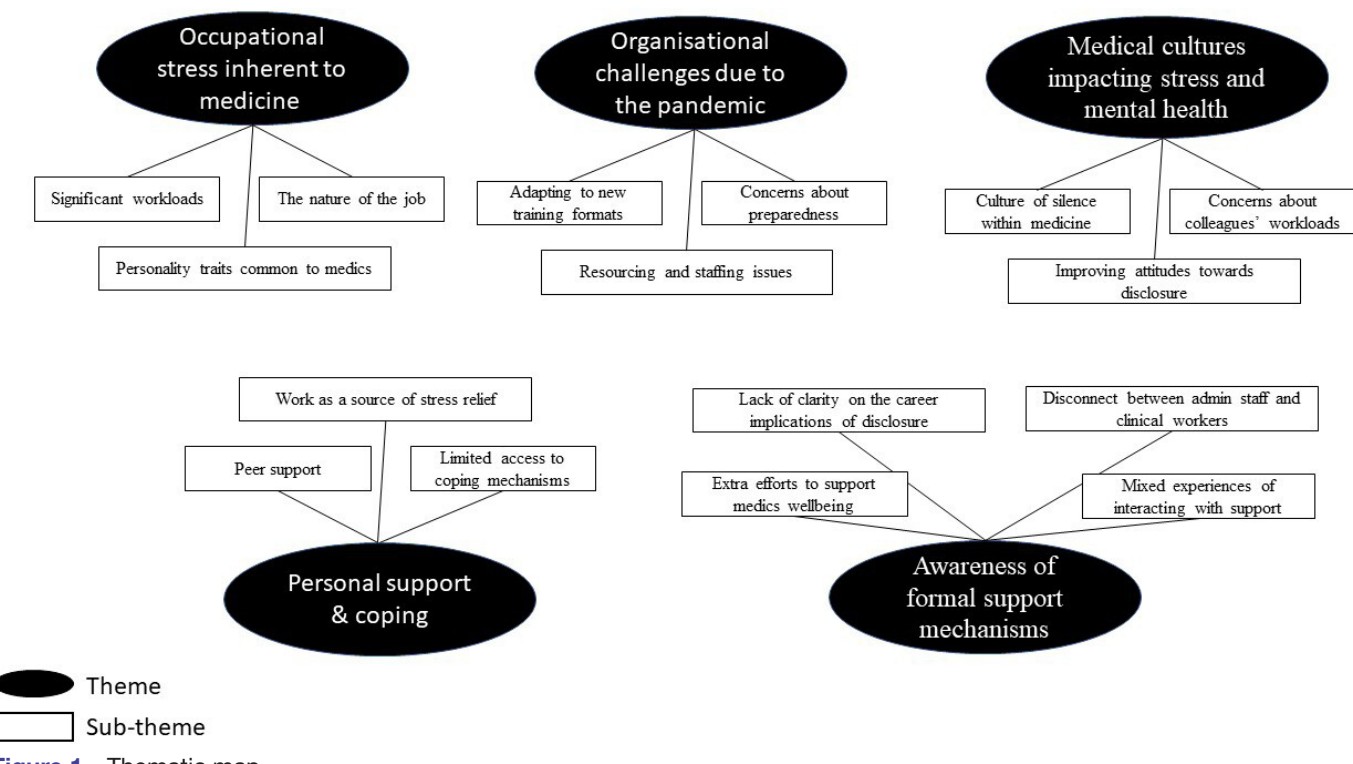

**Figure 1** Thematic map.

as Foundation Year 1 (FY1), while two were in their third year (FY3). The participants were working across the UK; 12 were male and the majority (13/15) were of a white British ethnic background (table 1).

## Occupational stress inherent to medicine

Sizeable workloads in medicine were a significant stressor among participants and many reported an increased prevalence of symptoms of depression and anxiety in the build-up to exams. Some participants felt that their work commitments were more excessive than other degree subjects or occupational roles, and that this might make them more susceptible to poor mental health.

> I lived with non-medics this year and I was quite surprised by how little they did (…), I think when you have to turn up to placement 9–5 every day… I think that pressure drains you and you do get worn down by it (#3, 5th year student).

All participants stated that practising medicine can be emotionally challenging, as reflected below from a final year medical student.

> When you're looking at the clinical side, I do think it is particularly challenging, for example when you're put in situations where you're watching someone die or trying to save someone's life (#2, 5th year student).

Moreover, participants remarked that personality traits common to medics, such as competitiveness, perfectionism, self-criticism and a tendency towards workaholism, can be maladaptive in terms of their well-being.

> The people that medicine attracts can be perfectionists where they might be more vulnerable to negative experiences and working in medicine comes with its fair share of those (#7, 4th year student).

Participants explained that you have to be resilient to practise medicine, to cope with the workload and the emotional aspects of the job. In some cases, participants alluded to a socialisation process which begins on entry to medical school, where students must claim or develop the resilience that the profession requires.

> I do think we are quite a resilient group of people, because you kind of have to be, and it's something that just happens to you when you're in medical school and you start working (#4, FY1).

## Organisational challenges associated with the COVID-19 pandemic

A majority of medical students suggested that changes to academic structure had compounded aspects of their education as well as their mental burden during the pandemic. Participants explained how online learning mechanisms provided fewer opportunities for social interaction, led to feelings of frustration and decreased their motivation to study.

> I much prefer face-to-face because there's much more interaction and it's so much easier to engage, and so much more sociable. You kind of lose motivation when you're just staring at a screen all day in a room by yourself (#2, 5th year student).

Concerns around preparedness were described, specifically where aspects of education had been impaired or missed altogether. This point is particularly salient for the subcohort of junior doctors who graduated early to offer their services during the pandemic.

> I've still never worked in the Emergency Department (ED), I've never even been there as a medical student, so when it eventually comes to me being in ED, I'll definitely feel less prepared in that regard (#5, FY1).

Despite this, several junior doctors suggested that even without the pandemic, they would have felt underprepared for the step up from medical school to a qualified doctor.

> You get to the point in medical school where you've done your exams and you're going to be as ready as you will be (#4, FY1).

Another source of stress, which particularly pertained to junior doctors, was resourcing problems within healthcare. For example, staffing problems left junior doctors feeling overburdened which compounded their experiences of work stress and impacted on the quality of care they could provide.

> The consultant came back and kind of said… you know, 'Have you done this? Have you done this? Has this happened?' and I was just oblivious to it. It was a particularly understaffed ward in a time when the hospital was understaffed (#4, FY1).

In some cases, a shortage of personal protective equipment led to concerns around risk of infection and personal health and safety. Moreover, participants suggested that a lack of resources including ventilators and hospital beds had created intensely stressful situations, where staff would regularly be making life and death decisions about who should receive care.

> That's quite hard knowing that you're sort of holding off optimal treatment for a patient because you don't have beds. Yeah, that was horrible actually, because then equally you've then got families, saying 'well, why aren't we escalating?' (#6, FY1).

## Medical cultures surrounding work-stress and mental health

Participants discussed cultural norms regarding occupational stress and mental health within the health service. A culture where healthcare workers' schedules and work–life balance were not prioritised was noted. Junior doctors explained that it was common to be asked to stay late or work on your day off at short notice. Participants

described experiencing pressure and guilt when being asked to work extra hours and suggested that it was particularly difficult to decline requests during the pandemic.

It's a really difficult one because you just feel like you should, you've got an obligation to do it, and it kind of became the attitude during the pandemic that you weren't doing anything anyway (#4, FY1).

In some cases, participants suggested that medical cultures can block access to mental health services and can perpetuate presenteeism behaviours. For example, some participants felt that it may be particularly difficult to disclose a mental health problem while practising medicine, due to stigmatisation. Specifically, concerns were raised that peers, tutors or supervisors may perceive them as being mentally weak, or lacking the resilient personality required to practice medicine, if they were to take time off for mental health difficulties.

Anyone that works does this, but particularly people that work in the NHS, you feel you have a duty to go to work, you know even if you're at death's door you go into work (#4, FY1).

.... I was worried that you know, if they do think that I'm not mentally 100% then I can't be doing this, because I can't handle the stress (#13, 4th year student).

All of the participants stated that attitudes towards mental health and disclosure have improved in recent times and this reflected broader perceptions and public awareness of the importance of good mental health. Moreover, participants agreed that recognition of mental health had acutely improved during the pandemic, as evidenced below.

The general consensus is that the kind of historic or traditional view of that is kind of 'pick yourself and get on with it', whereas I think over the last 12 months, but also over the last decade, there's definitely been a shift in acknowledging that health care professionals need to take care of themselves first and foremost (#1, 5th year student).

A number of participants indicated that they would not feel comfortable approaching senior doctors within their hospitals for mental health support. Despite this, interactions with senior doctors were generally positive and participants felt that they were helpful and accessible.

All through medical school you're taught about these really scary surgeons and consultants that won't take your answers but, on the whole, they've all been really supportive (#6, FY1).

Junior doctors worried about burdening colleagues with extra work if they were to take time off, mainly due to significant staff shortages during the pandemic. Despite this, medical students and junior doctors adopted an altruistic perspective, whereby they would feel guilty for adding to their colleagues' workloads but noted that

they would be happy to pick up the extra work if their colleagues needed time off for their mental well-being.

That's the good thing about being an FY1, you have such good camaraderie, you know you would do that (pick up the extra work when a colleague is off) for your colleagues, there's a real sense of that amongst the workforce (#4, FY1).

### Personal support and coping during the pandemic

The importance of peer support was unanimously felt by all participants. Many relied on their peers for informal mental health support and to problem solve difficult situations. A majority of the participants also lived with their coursemates or colleagues, which provided them with additional opportunities to open up about their problems.

We kind of do it without being labelled a support network, so we all come home from placement and will be like, 'I saw this thing today', and we tell the story, and we maybe don't consciously realise but we're saying it to get it off our chest (#14, 4th year student).

Beneficial work cultures, including camaraderie, humour and team dynamics, cultivated a sense of solidarity while working in challenging circumstances. Rewarding aspects of working in the medical profession were also highlighted. In particular, participants suggested that practising medicine had provided them with a sense of purpose and identified the psychological benefits associated with treating patients.

You are doing a job that is inherently helpful and rewarding and you feel good because of it and when something goes well, that buzz stays with you for a long time (#14, 4th year student).

Simultaneously, participants discussed how their usual methods for managing stress had been hindered by the COVID-19 pandemic. Many participants described feeling isolated and went extended periods without seeing their close friends and family, while several FY1s found it difficult to integrate into new work settings.

When lockdown was happening, I couldn't leave, and I think it got to the point where I was getting quite stressed about if my parents get sick, or my grandparents got sick, I wouldn't be there (#15, FY3).

### Awareness of formal support mechanisms

Participants felt that occupational health services had made extra efforts to support their well-being during COVID-19. Extra email communications, signposting to mental health services, well-being posters around hospitals and psychoeducational talks were some identified methods.

During the pandemic there's been a real focus on mental health for all members of staff, be that porters, nurses, doctors, auxiliary staff or healthcare

assistants, and there's been a real effort in reaching out to people (#4, FY1).

However, some participants felt that they would have benefited from receiving further support, such as more opportunities for rest and recovery, peer support and more accessible mental health services.

I think X university do it, they have mental health days where you can sort of, no questions asked, 'I'm not going in today', and you can just email them and say that, and you have a certain number throughout the year (#8, 5th year student).

Of the participants that had previously met with occupational services for their mental health, a majority suggested that they had not found it helpful. Participants felt that occupational services were more concerned about what this meant for their academic progression or work availability, rather than on offering them support. Some participants felt their negative experiences in dealing with occupational health services had discouraged them from accessing help in the future.

It was occupational health, and I had a meeting with her, and she was very, very condescending (…), I think that was quite a rare experience, but it's definitely tainted my view of disclosing mental health issues (#6, FY1).

Around half of the sample indicated that they would be concerned about career implications if they disclosed a mental health problem. Some participants felt this may affect their future employability. Moreover, a couple of participants indicated reluctance to disclose a mental health problem to their general practitioner, stating concerns about the confidentiality of their medical record.

It's something that you shouldn't be worried about because it shouldn't be too different to any other health condition, but it would worry me, I think it does worry me a little bit, so when I got the job with the NHS, I didn't put that down (#14, 4th year student).

While most participants felt supported by their clinical colleagues, tutors and peers, attitudes towards administrative staff were more variable. Participants reported feeling underappreciated and inadequately supported by management, suggesting that they lacked insight into the pressures involved in practicing medicine.

The senior doctors will try to help you out and give you advice, but not from the actual hospital management, rota co-ordinators, they have no clue, everything's like a number for them, it's almost like the business side of things so they don't know the pressures of actually working for the hospital (#11, FY1).

## DISCUSSION

Medical students and junior doctors discussed high levels of occupational stress which impacted on their well-being during the COVID-19 pandemic. Various stressors inherent to practising medicine, such as significant workloads and emotional challenges, were identified as well as organisational difficulties associated with the pandemic. Issues pertaining to resourcing, staffing, preparedness and adapting to new training formats exacerbated participants' work-stress during the pandemic. Extra efforts to support medical students and junior doctors during the pandemic were noted, however, participants also perceived barriers to accessing mental health support. These included a lack of clarity on their career implications, concerns about adding to colleagues' workloads, stigma in the profession and previous negative experiences with occupational health services.

Medical cultures which promote presenteeism and personality traits including perfectionism, self-criticism and competitiveness were viewed as influencing participants' work stress. Previously, a 'hidden curriculum' within medical training has been described, where trainees are indoctrinated with social norms that may be maladaptive.[22 24] These norms place medical trainees under pressure to accept insufficient working conditions,[24 25 49 50] or feel guilty for letting patients down, and burdening their colleagues with extra work if they themselves need time off.[6 51 52] At the same time, doctors receive validation from their peers for going above and beyond for their patients, producing an environment which sustains burn-out.[24]

Participants discussed factors that might prevent them from accessing help for their mental well-being, which echoed previous findings where medical cultures of secrecy and distrust were identified, as well as concerns around being perceived as weak.[53 54] Furthermore, participants worried about the career implications of disclosing a mental health problem to occupational and health services, indicating the need for improved clarity and transparency in this process. A previous study on depression, stigma and suicidal ideation in medical students revealed that 53% of respondents who reported depressive symptoms worried that revealing their illness would be risky for their career.[55] Participants also described negative experiences interacting with occupational services, which deterred them from accessing mental health support in the future.

All of the research participants recognised progress towards promoting and managing mental health within the medical profession. Participants described how attitudes towards mental health had improved acutely over the last 18 months, which may reflect extra efforts to support medical workers during the COVID-19 pandemic.[56] In line with previous findings,[57 58] the medical students and junior doctors noted the importance of peer support. However, a majority of participants indicated that the pandemic had limited their access to support and coping strategies which would usually benefit their well-being. Despite this, participants suggested that working in the

medical profession had been somewhat protective for their mental well-being during COVID-19, in that it is an inherently helpful, rewarding and sociable profession.

To our knowledge, no qualitative studies have simultaneously investigated the mental health of medical students and junior doctors during the COVID-19 pandemic. A further study strength is that an independent researcher carried out the interviews and analysis, adding an extra layer of privacy protection and strong reassurance of participant confidentiality. This study accessed a very unique cohort of medical students and junior doctors working in unprecedented times for the medical profession, therefore adding to the current sparse literature.[59] Despite this, the representativeness and transferability of the findings may be limited, as most participants were males and of a white British ethnicity. This is of particular concern as women, and ethnically diverse medics were reported to have suffered from higher rates of anxiety, depression and stress during the COVID-19 pandemic.[60–62] In addition, while snowball sampling proved a useful recruitment technique, selection and response biases cannot be excluded,[63] and the sample may not represent the experience of medics across the UK.

Protective measures should be implemented to minimise medical students' and junior doctors' occupational stress. These might include ensuring that workers only work their scheduled hours, encouraging more autonomy in shift allocation and consistently monitoring their well-being.[64] In particular, communications should be aimed towards leadership structures, to promote supportive and compassionate cultures, and to role model setting boundaries around work–life balance.[6 22] These practices should be in place from medical school onwards, when first year students begin to be socialised into the profession.[24] Finally, individuals who need to take time off should feel able to do so and be made aware of the support systems in place.[56]

The importance of peer support was clear in the findings. Consequently, interventions which foster peer support must be considered, and existing support from colleagues, supervisors and teams should be used.[22] Elsewhere, good practices have included 'buddying-up' schemes to reduce isolation and peer support groups where medics can discuss the emotional impact of their work.[56 65] Furthermore, the career implications of disclosing mental health problems must be clarified. Educational and training interventions may be adopted, which can encourage help seeking behaviours and boost medics' resilience.[64 66 67]

Given the exploratory nature of this research, larger qualitative and quantitative studies are required to determine the generalisability and transferability of the findings. In particular, studies investigating perceived differences in mental health problems by sociodemographic, educational and geographical factors are warranted. Future studies may benefit from exploring the views of educational, occupational and healthcare management staff, to add other perspectives on the factors affecting this cohort's well-being. Further validation of the findings could also be added using observational research in order to provide more impetus for quality improvement interventions to be implemented.

As we move away from the pressures of COVID-19 pandemic, the systemic and cultural issues within medicine remain highly relevant. This paper advocates for improvement to working conditions, more communication and connection within teams, and for cultural change, in medical education and in health services.

**Acknowledgements** The authors thank all the participants for sharing their experiences.

**Contributors** JT: conceived and designed the study; conducted the participant interviews, data analyses, drafted the manuscript and is the guarantor for the paper; ED: contributed to the interpretation of the results and data analysis and with critical revisions of the manuscript.

**Funding** ED acknowledges funding from the Medical Research Council (MC_UU_00022/2) and the Chief Scientist Office (SPHSU17).

**Competing interests** An initial sample of participants were recruited through personal contacts of the lead researcher and the remainder were recruited using snowball sampling. ED declares no competing interests.

**Patient and public involvement** Patients and/or the public were not involved in the design, or conduct, or reporting, or dissemination plans of this research.

**Patient consent for publication** Consent obtained directly from patient(s).

**Ethics approval** This study involves human participants and was approved by University of Glasgow, School of Social & Political Sciences Ethics Committee Reference #: PGT/SPS/2021/035/GLOB. Participants gave informed consent to participate in the study before taking part.

**Provenance and peer review** Not commissioned; externally peer reviewed.

**Data availability statement** No data are available.

**ORCID iDs**
James Tawse http://orcid.org/0000-0003-2022-6106
Evangelia Demou http://orcid.org/0000-0001-8616-525X

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
