## [Reviewer comments · BMJ Open]

ARTICLE DETAILS

TITLE (PROVISIONAL)	A qualitative study to explore UK medical students' and junior doctors' experiences of occupational stress and mental health during the COVID-19 pandemic
AUTHORS	Tawse, James; Demou, Evangelia

VERSION 1 – REVIEW

REVIEWER	Creese, Jennifer University of Leicester, Department of Health Sciences
REVIEW RETURNED	26-Jul-2022

GENERAL COMMENTS	Review Comments: “Medical students and junior doctors’ perceptions and experiences of occupational stress and mental health during the COVID-19 pandemic” (Manuscript ID bmjopen-2022-065639) Thank you very much for the opportunity to review this manuscript, which details a qualitative study of the experiences of medical students and junior doctors working in the UK during the COVID-19 pandemic. This is a crucially important topic, as the voices of doctors in training are often lost in narratives of healthcare work, and these more junior professionals less likely to speak up over working conditions and issues at work for a variety of reasons. Yet these participants will play a significant role in the rebuilt post-pandemic health system, and their retention is crucial to health workforce, so their insights are very necessary and this paper provides a good vehicle for bringing these to light. As a piece of scholarship, this study has valuable insights to make; however I believe it does need some substantial strengthening to be able to make a strong contribution both in terms of academic content and potential policy and practice impact. Further detailed 1. Is the research question or study objective clearly defined? No. To start with, I feel it suffers a little from being situated right in the middle of COVID and not being able to look beyond. It does not frame the situation of the pandemic clearly, either globally nor in the specific UK context, nor really look much at how the context before COVID shaped the situation in which the participants are working/studying or how the issues/findings might play out longer term after COVID. This is particularly important to consider if the study is to become part of the scholarly record of COVID-19 intelligence to be applied to future-proofing the health workforce, health system, medical education etc. It would benefit from some expansion in this regard: begin by establishing a stronger background – when did COVID come in, what was the UK response in terms of health system, society and education? While this is the BMJ, it still has an international audience, and providing the contextual background of what the UK situation was at the time of the study will increase understanding. Then you can introduce specifically what is known about junior doctor and medical student experiences within this context – but be aware
---

that what is missing from the introduction at the moment is a strong “So What” factor. Why is it important to consider the specific experiences of these cohorts? (This is where a longer-view framework would be helpful; it’s not just about COVID it’s about retaining and motivating the future of the profession in a period of long-term need) What does this study add that the other cited studies don’t? The aim to “develop a greater understanding of the work-stressors...explore their barriers to accessing care for mental health...and assess how educational and health services have adapted” is good, but why is it important to do this? What does it aim to achieve, even as an exploratory study what is the purpose of exploring this?

Take care with the introduction that the previously-published findings about stress, burnout and need for mental healthcare presented don’t overpower the findings this study will be coming up with (this should more rightly be in a discussion, where you can say “our finding of X also”) and in the introduction mention context (e.g. cancelled training, study in isolation, and being expedited to frontline service) but not findings. Also be clear what is a UK/NHS specific context here, as not all countries had medical students and junior doctors doing the same things as part of their pandemic response.

Finally, recommend a clear subsection/paragraph for defining, supported by literature, what the authors mean by “occupational stress”, “mental health”, “wellbeing” etc, whichever terms are to be consistently used throughout so readers can conceptualise what this covers (particularly where some terms e.g. wellbeing are complex: see eg <https://www.mdpi.com/1660-4601/18/4/2051> for a solid literature overview on wellbeing)

2. Is the abstract accurate, balanced and complete?

Yes, but consider the addition of the term “qualitative” somewhere here to make methodology/approach clear and increase findability/utility of the piece. (Perhaps “This study qualitatively explores the occupational experiences...”?)

3. Is the study design appropriate to answer the research question?

Yes, but this has not been adequately established by the authors. Two quantitative studies on the topic have been cited, but then a “paucity” of qualitative studies has been used to justify this study (none are cited, yet this study should respond to other qualitative pieces in this very journal on the same topic, namely

<https://bmjopen.bmj.com/content/11/5/e049437>

<https://bmjopen.bmj.com/content/11/12/e056122> and

<https://pubmed.ncbi.nlm.nih.gov/34373310/>). Importantly, no exploration has been made into why a qualitative study is needed, what it would add beyond these quantitative papers. What does qualitative methods literature say that defends why qualitative research can add more insight, nuance etc to the topic beyond the quantitative thinking? (see e.g.

<https://dx.doi.org/10.1097/MPG.0b013e3182a025d8>)

4. Are the methods described sufficiently to allow the study to be repeated?

No – the authors need to more strongly support their selection of methods with methodological literature. For example, a sample appears to have been used from the lead author’s networks; does this mean a convenience sample (i.e. they messaged all their contacts?) or purposively sampling (i.e. they messaged specific friends they wished to recruit?). In whichever case it is (be specific) how is this a suitable

method of recruitment, where has it been done in other studies to support your use of it here?

5. Are research ethics (e.g. participant consent, ethics approval) addressed appropriately?

Yes – but the line regarding institutional review board ethics permission should be connected with the authors' details of their own ethical practice efforts (page 8 lines 12-24). The authors' attention to ethical practice is commendable but needs to be more specific (e.g. what does "several ethical assurances" actually mean?). Consider reinforcing here with more methodological literature on applied researcher ethics in qualitative work (e.g. Antoni & Beer's chapter "Research impact as care: re-conceptualizing research impact from an ethics of care perspective" in 2019 Business Ethics and Care in Organizations <https://books.google.co.uk/books?id=JlrcDwAAQBAJ>)

6. Are the outcomes clearly defined?

No – the inclusion of a graphical representation of the themes was a good idea but the formatting of this is awkward and does not flow logically as it branches out in different directions/dimensions that do not align with clear themes. The existing thematic structure is probably too complex to adequately represent in this format. The three RQs are effectively themes, the 5 themes are sub-themes, and then there is a third level of sub-sub-themes below this! Consider stripping back the third level, and presenting across one horizontal plane rather than two left and one right. If this will not fit legibly consider tabulating the themes instead.

7. If statistics are used are they appropriate and described fully?

Not Applicable

8. Are the references up-to-date and appropriate?

Yes, though suggestions for additions made throughout review

9. Do the results address the research question or objective?

Yes – identified themes clearly derive from the research questions posed at the start.

10. Are they presented clearly?

Yes; sections clearly follow the themes identified in the start of the Results section (though these 5 main themes could be listed in text here as well as in the diagramme.) There is a lovely use of the participants' voices throughout here. The only thing I would caution is consistency in terminology between "mental" and "psychological" health (and certainly not "psychiatric" as this would be reserved for chronic mental illnesses); it would help if these terms were defined as per 1 above to frame the results.

11. Are the discussion and conclusions justified by the results

Yes – but while the discussion has connected results to similar findings from complementary studies, it misses the opportunity to unpack some of the factors behind these findings. For example, the finding is presented that "Medical cultures which promote presenteeism and personality traits including perfectionism, self-criticism, competitiveness and resilience were viewed as influencing factors on participants' work stress." However, this needs to be established that these are well-established medical cultural norms (with literature reference e.g. <https://www.sciencedirect.com/science/article/pii/S0008418219312797>) rather than just participants' views; that strengthens the case for something to be done about the problem.

	12. Are the study limitations discussed adequately? Yes – though recommend going back over limitations to talk about why these identified limitations are a problem. For example, yes the gender and ethnic makeup of participants doesn't lend itself to generalisability; but more than that, it's well established that the pandemic was tougher for both women (https://www.mdpi.com/2076-0760/10/2/43) and ethnically diverse healthcare workers (https://static.frontiersin.org/articles/10.3389/fmed.2022.930904/full). The conclusion begins well, with clear implications for policy and practice, and the directions for future research are good and practical. However, recommend one final paragraph restating what the research set out to do, what it found in summary and the "so what", why it's important, just to hit the paper home. 13. Is the supplementary reporting complete (e.g. trial registration; funding details; CONSORT, STROBE or PRISMA checklist)? Yes – SRQR checklist has been included, but it is worthwhile including in the text that this has been followed also (in the Methods section, with reference) 14. To the best of your knowledge is the paper free from concerns over publication ethics (e.g. plagiarism, redundant publication, undeclared conflicts of interest)? Yes 15. Is the standard of written English acceptable for publication? Yes
--	--

REVIEWER	Spiers, Johanna University of Birmingham, College of Medical and Dental Sciences
REVIEW RETURNED	28-Jul-2022

GENERAL COMMENTS	Thank you for giving me the opportunity to review this paper, which is clearly organised, well-written and about an important topic. I think that, with the following revisions, the paper will make an important contribution to the literature. I hope the amendments I've suggested don't feel overwhelming. While there are quite a few of them, I believe them all to be relatively minor. Well done on a great paper. Abstract: This is clearly written - the study objectives are laid out well and each section feels thorough and full. This is an important topic and a unique sample, having been captured at this difficult time in history. Well done. Introduction: This is another well-written section. You make a clear argument for why this topic is important given the conditions med students and junior doctors were working in at this time. One point I would make is that there has been some high-quality qualitative work on junior doctors working during the pandemic:
---

Spiers, J., Buszewicz, M., Chew-Graham, C., Dunning, A., Taylor, A. K., Gopfert, A., ... & Riley, R. (2021). What challenges did junior doctors face while working during the COVID-19 pandemic? A qualitative study. *BMJ open*, 11(12), e056122.

As well as other qualitative papers looking at NHS workers experiences during the pandemic more generally - for example:

Newman, K. L., Jevé, Y., & Majumder, P. (2022). Experiences and emotional strain of NHS frontline workers during the peak of the COVID-19 pandemic. *International Journal of Social Psychiatry*, 68(4), 783-790.

I would suggest citing these papers in your introduction and explaining how your paper builds on what is already in the literature.

If, as your thematic map (Fig. 1) suggests, you had three clear, distinctly worded research questions, I suggest presenting these as bullet points in the final para of your intro, to make it crystal clear to the reader what objectives you wanted to meet and how these questions fit into your thematic map.

Methods:

Your inclusion criteria are a little confusing. Was it your intention to include med students from the outset, or was this a later decision? Can you reword the first two sentences of your method section - and indeed the corresponding section of the abstract - to clarify this?

Did you not carry out face-to-face interviews because of the pandemic? If so, please make that clear.

The interviews seem rather short for a qualitative study on such an emotive topic. Can you comment on why this was?

The Braun and Clarke paper you cite is quite old now. B&C have published many more recent papers in which they develop and further explicate thematic analysis. Did you consult any of those papers? I would particularly suggest reading this paper:

Braun, V., & Clarke, V. (2019). Reflecting on reflexive thematic analysis. *Qualitative research in sport, exercise and health*, 11(4), 589-597.

Having read your results section, I feel that your themes are perhaps at the 'domain' level rather than the 'theme' level (as described in the above paper) as they are about a shared topic rather than shared meanings - they capture a diversity of meaning within related areas. Can you comment on this?

I see that you mention data saturation, a term which comes from grounded theory and refers to the conclusion of a process in which researchers analyse data simultaneously to interviewing, and use their early theories and concepts to theoretically sample subsequent participants and interviews so that those theories and

concepts can be tested and built upon. Achieving data saturation in a thematic analysis with only 15 participants seems quite ambitious. Can you say more to explain how you define saturation and how you determined that you had achieved it?

Results

Table 1 feels rather quantitative to me, and is a little hard to read. I'm not sure the percentages are meaningful in a qualitative study with a small sample. More typically, tables of participant demographics in qualitative studies are presented with participant ID numbers in the left-hand column and then further columns for other details - in this case, training level, place of work, gender ID and ethnicity. I've attached an example - unless you feel this compromises the anonymity of your participants, I would suggest presenting the table like this. If this would compromise confidentiality, I would still suggest reformatting this table in some way to make it less number-focused and more readable.

I also feel that your thematic map needs a little further explanation. Where did these research questions come from? Are these the RQs that informed your study, or questions you asked participants in your interviews? The third question (have med students and JDs received an adequate level of support to manage their work-stress and mental health challenges during the pandemic) doesn't seem to match up to the third objective you currently list in your intro (how educational and health systems have adapted their systems to support students and JDs). The question in the thematic map and abstract focuses on the participants - the wording in the intro focuses on the systems. Can you clarify?

An explanation of what the various straight and dotted lines in the thematic map indicate would be useful too.

Theme 1: I would like to see the claim about high workloads supported with a quote from one of the junior doctors as well as the quote from the med student.

I suggest keeping quotes next to the comments about those quotes, as this helps the reader to follow your train of thought. So I would place the quote from #2 after the first line of para 3 on page 9, and then move onto the comment and quote about resilience. This is an issue at other points in the results section, so watch out for other places where you could clarify which comments 'belongs' to which quote.

It would also be good to see a quote to support the claim that personality traits of medics could add to occupational stress.

I wonder if you can unpack the quote about resilience a little further. Note that the participant says medics are resilient 'because we have to be' and that resilience 'just happens to you'. This data is ambiguous, perhaps suggesting a culture in which it is unacceptable NOT to claim resilience, rather than one in which genuine resilience is nurtured. What do you think?

	Theme 2: It would be good to see a quote which evidences your point about the missed education being especially hard for the JDs who graduated early - at present, I think you have two quotes from participants who felt they wouldn't have been ready no matter what? If I'm wrong, please clarify which quote comes from one of the early graduates and provide more context. Theme 3: This is a well-organised and well-evidenced theme, well done. Theme 4: While this is another well-organised and well-written theme, it is striking to me that there are so many more positive quotes than quotes about the challenges of being supported and coping during COVID-19 (especially given your claim in the discussion that most participants felt there was less support during the pandemic). Was it the case that participants were more positive than negative about support and coping during this time? (Fantastic, if so!) Or does this balance need redressing a little? Theme 5: The first quote in this theme only seems to evidence your initial point, ie that extra efforts were made to support staff. Can you move this quote up so it appears underneath that comment and then also include a quote to illustrate your second very important point about the need for further support? The data in this theme is really strong and presents a worrying picture of mental health support in the NHS. Taken in conjunction with the quote in theme 1 where a participant says that doctors 'have to' be resilient, this is further evidence for the toxic culture that my colleagues and I have reported in various papers: Riley, R., Spiers, J., Buszewicz, M., Taylor, A. K., Thornton, G., & Chew-Graham, C. A. (2018). What are the sources of stress and distress for general practitioners working in England? A qualitative study. BMJ open, 8(1), e017361. Riley, R., Buszewicz, M., Kokab, F., Teoh, K., Gopfert, A., Taylor, A. K., ... & Chew-Graham, C. (2021). Sources of work-related psychological distress experienced by UK-wide foundation and junior doctors: a qualitative study. BMJ open, 11(6), e043521. Discussion You've said in the discussion that 'a majority of participants indicated that the pandemic had limited their access to support networks and coping strategies'. However, the findings suggest that there was more support available, plus most quotes in theme 4 suggest that participants were able to find ways to cope and be supported. This currently feels like a bit of a mismatch. Can you clarify? I don't agree with your second limitation, ie that the participants would find it hard to contextualise stress outside of the pandemic - your study is about medical students' and JDs' stress during the pandemic. I would suggest that this subject matter precludes any contextualisation away from COVID-19. I recommend deleting this limitation as I think you're doing yourself a disservice here.
--	---

	While the protective measures you've suggested all sound good in theory, I wonder if you can comment on how realistic they are in practice? This paper might be useful to you here: Riley, R., Kokab, F., Buszewicz, M., Gopfert, A., Van Hove, M., Taylor, A. K., ... & Chew-Graham, C. (2021). Protective factors and sources of support in the workplace as experienced by UK foundation and junior doctors: a qualitative study. BMJ open, 11(6), e045588. A few proofreading points: This paper, while mostly well-written, contains some awkward-sounding sentences and grammar/punctuation errors. I would suggest running the paper through the free online version of Grammarly to iron those issues out - but here are some issues I've spotted. You've got a stray comma after 'junior doctors' in the first line of the second bullet point of the article summary. I would also suggest breaking this second bullet point into two sentences: '...training opportunities. It provides...' There are several other misplaced or missing commas in this paper, so do look out for those. This sentence: 'Prior to arranging interviews, participants were made aware of the requirements and purposes of the research, as well as several ethical assurances, so that they could make an informed decision on participation' reads a little strangely. I would suggest '...of the research, and were given assurances about the ethics of the study so that they could make an informed decision about participation.'
--	--

REVIEWER	Waechter, Randall St George's University
REVIEW RETURNED	31-Aug-2022

GENERAL COMMENTS	This is an important topic of exploration given the pre-existing high rates of stress, burnout, and related mental illnesses among medical students and junior physicians (as well as physicians as a whole, compared to the non-physician population) prior to the COVID-19 pandemic. Medical educators and healthcare administrators need to have in-depth knowledge of how the pandemic impacted medical students and physicians so that effective support systems can be established for future health emergencies, whether at the local, regional, national, or international level. This manuscript provides an important contribution to the base of knowledge and I commend the authors for pursuing it. It was particularly enlightening to see the responses from the participants regarding stigmatization around mental illness, which confirms the amount of education and awareness work yet to be done within the healthcare field, let alone among the general population.
---

	There are a few limitations to the study, which must be taken into account when interpreting and applying the results.  1. The study aims was to "to develop a greater understanding of the work-stressors experienced by medical students and junior doctors during the pandemic....and assess how educational and health services have adapted their systems to support medical students and junior doctors during the COVID-19 pandemic". I suggest the authors add "in the UK" to clarify for the reader that this study applies specifically to that country. 2. As per the above, I also suggest the author's add: "in the UK" to the title of the manuscript. 3. The sample size is relatively small and the sampling technique is not randomized. The authors state that selection response bias cannot be ruled out, but perhaps a more explicit statement about the sample not necessarily representing the experience of physicians across the UK is warranted. 4. The authors provide suggestions for increasing access to occupational health care for medical students and junior doctors. There is good evidence that changes to the way medical training is provided is also helpful in reducing burnout and stress among students (e.g., Bloodgood RA, Short JG, Jackson JM, Martindale JR. A change to pass/fail grading in the first two years at one medical school results in improved psychological well-being. Academic medicine : journal of the Association of American Medical Colleges. 2009;84(5):655-62; Reed DA, Shanafelt TD, Satele DW, Power DV, Eacker A, Harper W, et al. Relationship of pass/fail grading and curriculum structure with well-being among preclinical medical students: a multi-institutional study. Academic medicine : journal of the Association of American Medical Colleges. 2011;86(11):1367-73; Rohe DE, Barrier PA, Clark MM, Cook DA, Vickers KS, Decker PA. The benefits of pass-fail grading on stress, mood, and group cohesion in medical students. Mayo Clinic proceedings. 2006;81(11):1443-8). There is also evidence that behavioral programs instituted in medical schools can provide students with the skills to support resilience and the authors may want to mention this (e.g., Yiu V. Supporting the well-being of medical students. CMAJ: Canadian Medical Association Journal. 2005;172(7):889-890).
--	--

VERSION 1 – AUTHOR RESPONSE

REVIEWER 1

Dr. Jennifer Creese, University of Leicester Comments to the Author:

1a. Is the research question or study objective clearly defined?

No. To start with, I feel it suffers a little from being situated right in the middle of COVID and not being able to look beyond. It does not frame the situation of the pandemic clearly, either globally nor in the specific UK context, nor really look much at how the context before COVID shaped the situation in which the participants are working/studying or how the issues/findings might play out longer term after COVID. This is particularly important to consider if the study is to become part of the scholarly record of COVID-19 intelligence to be applied to future-proofing the health workforce, health system, medical

education etc. It would benefit from some expansion in this regard: begin by establishing a stronger background – when did COVID come in, what was the UK response in terms of health system, society and education? While this is the BMJ, it still has an international audience, and providing the contextual background of what the UK situation was at the time of the study will increase understanding.

Author response: We thank the reviewer for their comments. We have amended the introduction to ensure we address the above comments and suggestions. Our introduction now starts with a paragraph on the COVID-19 pandemic, followed by two paragraphs on the impacts to medical students and junior doctors, and then providing contextual background of what the UK healthcare situation was and why a qualitative study on the health and wellbeing of medical students and junior doctors was needed.

1b. Then you can introduce specifically what is known about junior doctor and medical student experiences within this context – but be aware that what is missing from the introduction at the moment is a strong “So What” factor. Why is it important to consider the specific experiences of these cohorts? (This is where a longer-view framework would be helpful; it’s not just about COVID it’s about retaining and motivating the future of the profession in a period of long-term need) What does this study add that the other cited studies don’t? The aim to “develop a greater understanding of the work-stressors...explore their barriers to accessing care for mental health...and assess how educational and health services have adapted” is good, but why is it important to do this? What does it aim to achieve, even as an exploratory study what is the purpose of exploring this?

Author response: Please see our revised introduction as per the above comment. Specifically for this comment, we have strengthened our ‘so what’ by providing more justification as to why this research is needed and its potential implications for policy and practice. The amended section exploring these potential impacts now reads (page 5; paragraph 3):

The mental health burden of healthcare workers and medical students working during the pandemic has been the subject of interest in quantitative research studies ^{12 26-28}; but there has been less qualitative research on junior doctors ^{6 13 29-33} and medical students in particular ³⁴. As such, this paper can offer valuable insights into these groups’ relative stress, mental wellbeing and access to support during this difficult transitional period from student to doctor ³⁵, and in unprecedented circumstances. This qualitative study captures the voices of these groups at a distinct and testing time, amongst rising rates of COVID-19 in the summer of 2021 and the long-term impact of COVID-19 on healthcare workers’ wellbeing remains unclear ^{36 37}. Consequently, continued efforts to protect medical students’ and junior doctors’ wellbeing are warranted. In doing so, retention issues may be addressed, which threaten patient safety, and the future of health services ²⁵. This study aimed to explore medical students’ and junior doctors’ experiences of occupational stress and mental health during the COVID-19 pandemic in the UK. Specifically, this study addressed the following three research questions.

1c. Take care with the introduction that the previously-published findings about stress, burnout and need for mental healthcare presented don’t overpower the findings this study will be coming up with (this should more rightly be in a discussion, where you can say “our finding of X agrees with broader findings that medical students experienced X also”) and in the introduction mention context (e.g. cancelled training, study in isolation, and being expedited to frontline service) but not findings. Also be clear what is a UK/NHS specific context here, as not all countries had medical students and junior doctors doing the same things as part of their pandemic response.

Author response: Please see our responses to the comments 1a-c above comments, relating to the changes to our introduction. Furthermore, we have clarified both in the title and in the introduction when referring to the changes in the education and working conditions for medical students and junior doctors, that this is related to the UK context mainly.

1d. Finally, recommend a clear subsection/paragraph for defining, supported by literature, what the authors mean by “occupational stress”, “mental health”, “wellbeing” etc, whichever terms are to be consistently used throughout so readers can conceptualise what this covers (particularly where some terms e.g. wellbeing are complex: see eg <https://eu-west-1.protection.sophos.com?d=mdpi.com&u=aHR0cHM6Ly93d3cubWRwaS5jb20vMTY2MC00NjAxLzE4LzQvMjA1MQ==&i=NjlzZWU0ODkwYjVmMWMxZDgwMmMwMjRm&t=enNOWGFwV1I2cm45QUJ4QlZ4UIYxMwWt5UFBLR0gwQmIBR2xwU1I2ckc3WT0=&h=fd53f56f94354779a7a5c26e40a32550> for a solid literature overview on wellbeing).

Author response: We appreciate and acknowledge the usefulness of such a section with key definitions of terms that are used. However, given that the terms we use that are related to health are well described terms in the mental and occupational health literature, and our restrictions to expand due to the word count, we feel that it is better to opt to use our word count for presenting our primary data. We have however, made sure that terms that may not be widely and/or internationally known, such as junior doctor, are more clearly defined in the paper, and that all terms are used consistently (see also Reviewer 1 comment 10). For instance, we have defined now in our Methods Section how a Junior Doctor is defined (page 6; paragraph 3):

For the present study, ‘junior doctor’ refers to a doctor who graduated from medical school in the last five years, and who’s currently enrolled on a medical training programme.

2. Is the abstract accurate, balanced and complete?

Yes, but consider the addition of the term “qualitative” somewhere here to make methodology/approach clear and increase findability/utility of the piece. (Perhaps “This study qualitatively explores the occupational experiences...”?)

Author response: Agreed and changed. The first sentence of our abstract now reads:

“ Objectives: This qualitative study aimed to explore the occupational experiences of medical students and junior doctors working during the COVID-19 pandemic....”

3. Is the study design appropriate to answer the research question?

Yes, but this has not been adequately established by the authors. Two quantitative studies on the topic have been cited, but then a “paucity” of qualitative studies has been used to justify this study (none are cited, yet this study should respond to other qualitative pieces in this very journal on the same topic, namely <https://eu-west-1.protection.sophos.com?d=bmj.com&u=aHR0cHM6Ly9ibWpvcGVuLmJtai5jb20vY29udGVudC8xMS81L2UwNDk0Mzc=&i=NjlzZWU0ODkwYjVmMWMxZDgwMmMwMjRm&t=TK5PK2s3VXYxNkJFejZ1YmZCU01ZY2dxc0JTATdCWVBSbzZicExXT0ICOD0=&h=fd53f56f94354779a7a5c26e40a32550> <https://eu-west-1.protection.sophos.com?d=bmj.com&u=aHR0cHM6Ly9ibWpvcGVuLmJtai5jb20vY29udGVudC8xMS8xMi9IMDU2MTIy&i=NjlzZWU0ODkwYjVmMWMxZDgwMmMwMjRm&t=cCtkbm5VK1VwSUt4OVRtMFBTWk0zY0x1N0hwUDRPMjRjNWJvaThHSGhabz0=&h=fd53f56f94354779a7a5c26e40a32550> and <https://eu-west-1.protection.sophos.com?d=nih.gov&u=aHR0cHM6Ly9wdWJtZWQubmNiaS5ubG0ubml0Lmdvdi8zNDM3MzIxMC8=&i=NjlzZWU0ODkwYjVmMWMxZDgwMmMwMjRm&t=WFZCeDVBGRtBZFMyeEFWQjNYUXdHL2tKa1BZdTdENIRBOEE0dFhmU20xVT0=&h=fd53f56f94354779a7a5c26e40a32550>).

Importantly, no exploration has been made into why a qualitative study is needed, what it would add beyond these quantitative papers. What does qualitative methods literature say that defends why qualitative research can add more insight, nuance etc to the topic beyond the quantitative thinking? (see e.g. <https://eu-west-1.protection.sophos.com?d=doi.org&u=aHR0cHM6Ly9keC5kb2kub3JnLzEwLjEwOTcvTVBHLjBiMDE>

<https://eu-west-1.protection.sophos.com?d=nih.gov&u=aHR0cHM6Ly9wdWJtZWQubmNiaS5ubG0ubml0Lmdvdi8zNDM3MzIxMC8=&i=NjlzZWU0ODkwYjVmMWMxZDgwMmMwMjRm&t=WFZCeDVBGRtBZFMyeEFWQjNYUXdHL2tKa1BZdTdENIRBOEE0dFhmU20xVT0=&h=fd53f56f94354779a7a5c26e40a32550>).

Importantly, no exploration has been made into why a qualitative study is needed, what it would add beyond these quantitative papers. What does qualitative methods literature say that defends why qualitative research can add more insight, nuance etc to the topic beyond the quantitative thinking? (see e.g. <https://eu-west-1.protection.sophos.com?d=doi.org&u=aHR0cHM6Ly9keC5kb2kub3JnLzEwLjEwOTcvTVBHLjBiMDE>

<https://eu-west-1.protection.sophos.com?d=doi.org&u=aHR0cHM6Ly9keC5kb2kub3JnLzEwLjEwOTcvTVBHLjBiMDE>

[zZTMxODJhMDI1ZDg=&i=NjlzZWU0ODkwYjVmMWMxZDgwMmMwMjRm&t=Z1dyb1hGM9NaDhMNU9XR2dlc3BXVmZmRHdhSEVYWUVZTjB3Zzd0MHR1MD0=&h=fd53f56f94354779a7a5c26e40a32550\)](https://doi.org/10.1111/j.1365-2648.2021.09435.x)

Author response: Thank you for providing these references. Due to the fact that the timing these studies were published coincided with our paper being prepared for submission, they were missed. We have now referenced this work and also clarified how our study further contributes to the literature, and especially how to our knowledge there is no other study examining the mental health of final year medical students (please see our revised introduction and response to comments 1a-d and 2 above) and specifically on the importance of the qualitative methods used (page 5; paragraph 3). *The mental health burden of healthcare workers and medical students working during the pandemic has been the subject of interest in quantitative research studies ^{12 26-28}; but there has been less qualitative research on junior doctors ^{6 13 29-33} and medical students in particular ³⁴. As such, this paper can offer valuable insights into these groups' relative stress, mental wellbeing and access to support during this difficult transitional period from student to doctor ³⁵, and in unprecedented circumstances. This qualitative study captures the voices of these groups at a distinct and testing time, amongst rising rates of COVID-19 in the summer of 2021 and the long-term impact of COVID-19 on healthcare workers' wellbeing remains unclear ^{36 37}. Consequently, continued efforts to protect medical students' and junior doctors' wellbeing are warranted. In doing so, retention issues may be addressed, which threaten patient safety, and the future of health services ²⁵. This study aimed to explore medical students' and junior doctors' experiences of occupational stress and mental health during the COVID-19 pandemic in the UK. Specifically, this study addressed the following three research questions.*

4. Are the methods described sufficiently to allow the study to be repeated?

No – the authors need to more strongly support their selection of methods with methodological literature. For example, a sample appears to have been used from the lead author's networks; does this mean a convenience sample (i.e. they messaged all their contacts?) or purposively sampling (i.e. they messaged specific friends they wished to recruit?). In whichever case it is (be specific) how is this a suitable method of recruitment, where has it been done in other studies to support your use of it here?

Author response: We have clarified that we have used a combination of sampling methods, initially purposeful sampling followed by recruitment via snowballing. This is common in qualitative research and we have also referenced other studies using similar and combined techniques. The revised section now reads (page 6; paragraph 3): *A combination of sampling strategies were used to recruit participants ³⁸. Initially, purposeful sampling of participants was implemented, and participants were recruited through personal contacts and acquaintances of the lead researcher. Following this, participants were recruited using snowball sampling ^{39 40}.*

5. Are research ethics (e.g. participant consent, ethics approval) addressed appropriately?

Yes – but the line regarding institutional review board ethics permission should be connected with the authors' details of their own ethical practice efforts (page 8 lines 12-24). The authors' attention to ethical practice is commendable but needs to be more specific (e.g. what does "several ethical assurances" actually mean?). Consider reinforcing here with more methodological literature on applied researcher ethics in qualitative work (e.g. Antoni & Beer's chapter "Research impact as care: re-conceptualizing research impact from an ethics of care perspective" in 2019 Business Ethics and Care in Organizations [https://eu-west-](https://eu-west-1.protection.sophos.com?d=google.co.uk&u=aHR0cHM6Ly9ib29rcy5nb29nbGUuY28udWsvYm9va3)

[1.protection.sophos.com?d=google.co.uk&u=aHR0cHM6Ly9ib29rcy5nb29nbGUuY28udWsvYm9va3](https://eu-west-1.protection.sophos.com?d=google.co.uk&u=aHR0cHM6Ly9ib29rcy5nb29nbGUuY28udWsvYm9va3)

[M_aWQ9SklyQ0R3QUFRQkFK&i=NjlzZWU0ODkwYjVmMWMxZDgwMmMwMjRm&t=QWxscFZuT1hBRnRtRHNST2l2emhKOUR6aDZzWWd4TVFrZXh5cidOeTE4RT0=&h=fd53f56f94354779a7a5c26e40a32550\)](https://doi.org/10.1111/j.1469-7610.2014.01411.x)

Author response: Thank you for your comments and for the recommendation of Antoni and Beer's work. We have amended the ethics section to be more specific and have justified our practices by citing methodological literature. The revised section now reads (page 6; paragraph 4):

Once sampled, participants were contacted to arrange one-to-one interviews and interviews were arranged for a time that suited the participants⁴¹. Before starting the interview, participants were made aware of the requirements and purposes of the research, the identity of the researcher, and how the results would be used⁴². Participants were reminded of their ethical rights surrounding confidentiality, data protection and their right to withdraw at any point⁴³. A safety plan was in place, to ensure that support could be provided in the event of a concerning disclosure⁴⁴. Following this, participants made an informed decision on their participation.

6. Are the outcomes clearly defined?

No – the inclusion of a graphical representation of the themes was a good idea but the formatting of this is awkward and does not flow logically as it branches out in different directions/dimensions that do not align with clear themes. The existing thematic structure is probably too complex to adequately represent in this format. The three RQs are effectively themes, the 5 themes are sub-themes, and then there is a third level of sub-sub-themes below this! Consider stripping back the third level, and presenting across one horizontal plane rather than two left and one right. If this will not fit legibly consider tabulating the themes instead.

Author response: Thank you for this comment. We appreciate that the presentation of our thematic map may have been too 'busy' and inserting our research questions within the map seems to have caused confusion. The research questions were guides to develop the interview schedule and were not questions asked directly to the participants. From our thematic analysis, there were five themes i.e. patterns of shared meaning- that emerged. These were the central concepts that illustrated patterns across the data and across the questions that were asked of the participants. We have now revised our thematic map- and removed the presentation of our research questions which were there for illustrative purposes as to how our themes 'mapped' onto the research questions of the study. We have now only presented our themes and the subthemes in a simpler and clearer way.

7. If statistics are used, are they appropriate and described fully?

Not Applicable

8. Are the references up-to-date and appropriate?

Yes, though suggestions for additions made throughout review

Author response: Thank you we have made all the appropriate changes.

9. Do the results address the research question or objective?

Yes – identified themes clearly derive from the research questions posed at the start.

Author response: Thank you for your consideration.

10. Are they presented clearly?

Yes, sections clearly follow the themes identified in the start of the Results section (though these 5 main themes could be listed in text here as well as in the diagram.) There is a lovely use of the participants' voices throughout here. The only thing I would caution is consistency in terminology between "mental" and "psychological" health (and certainly not "psychiatric" as this would be reserved for chronic mental illnesses); it would help if these terms were defined as per 1 above to frame the results.

Author response: We thank the reviewer for the positive comments and suggestions. We have checked throughout, and amended where necessary, to ensure we are consistent with the terminology.

11. Are the discussion and conclusions justified by the results

Yes – but while the discussion has connected results to similar findings from complementary studies, it misses the opportunity to unpack some of the factors behind these findings. For example, the finding is presented that “ Medical cultures which promote presenteeism and personality traits including perfectionism, self-criticism, competitiveness and resilience were viewed as influencing factors on participants’ work stress.” However, this needs to be established that these are well-established medical cultural norms (with literature reference e.g. <https://eu-west-1.protection.sophos.com?d=sciedirect.com&u=aHR0cHM6Ly93d3cuc2NpZW5jZWVpcmlVjC5jb20vc2NpZW5jZS9hcnRpY2xlL3BpaS9TMDAwODQxODIxOTMxMjc5Nw==&i=NjZlZWU0ODkwYjVmMWMxZDgwMmMwMjRm&t=aFBpRjQ3eWFPbnYxMVhXOTI3Nm9CNC9wcHJldk1aXhDYnE3TkFnkl4TT0=&h=fd53f56f94354779a7a5c26e40a32550>) rather than just participants’ views; that strengthens the case for something to be done about the problem.

Author response: Thank you for this comment. We have included a section which explores medical cultures around work in more depth. This section has been included below and can be found on page 19 (paragraph 2) of the revised manuscript.

Medical cultures which promote presenteeism and personality traits including perfectionism, self-criticism, and competitiveness were viewed as influencing participants’ work stress. Previously, a ‘hidden curriculum’ within medical training has been described, where trainees are indoctrinated with social norms that may be maladaptive^{22 24}. These norms place medical trainees under pressure to accept insufficient working conditions^{24 25 49 50}, or feel guilty for letting patients down, and burdening their colleagues with extra work if they themselves need time off^{6 51 52}. At the same time, doctors receive validation from their peers for going above and beyond for their patients, producing an environment which sustains burnout²⁴.

12. Are the study limitations discussed adequately?

Yes – though recommend going back over limitations to talk about why these identified limitations are a problem. For example, yes the gender and ethnic makeup of participants doesn’t lend itself to generalisability; but more than that, it’s well established that the pandemic was tougher for both women (<https://eu-west-1.protection.sophos.com?d=mdpi.com&u=aHR0cHM6Ly93d3cubWRwaS5jb20vMjA3Ni0wNzYwLzEwLzlvNDM=&i=NjZlZWU0ODkwYjVmMWMxZDgwMmMwMjRm&t=RCtKMVVITUY3bW16WTJDUFFPZVVZTkt6R0s4OGNoSC80c0I2T0o5aC85RT0=&h=fd53f56f94354779a7a5c26e40a32550>) and ethnically diverse healthcare workers (<https://eu-west-1.protection.sophos.com?d=frontiersn.org&u=aHR0cHM6Ly9zdGF0aWMuZnJvbnRpZXJzaW4ub3JnL2FydGlibGVzLzEwLjMzODkvZm1lZC4yMDIyLjkzMDkwNC9mdWxs&i=NjZlZWU0ODkwYjVmMWMxZDgwMmMwMjRm&t=QWNHY3h1ZDJJWTg3N25hRFIwVmdWMDJKYmZ6ZTdXWE9ubzUzVIJYcmR0UT0=&h=fd53f56f94354779a7a5c26e40a32550>).

The conclusion begins well, with clear implications for policy and practice, and the directions for future research are good and practical. However, recommend one final paragraph restating what the research set out to do, what it found in summary and the “so what”, why it’s important, just to hit the paper home.

Author response: Thank you for these comments, we have now amended our limitations to further discuss why our sample demographics may be an issue (page 21, paragraph 1). We have also included a brief final ‘so what’ section to conclude the paper, attached below.

As we move away from the pressures of COVID-19 pandemic, the systemic and cultural issues within medicine remain highly relevant. This paper advocates for improvement to working conditions, more communication and connection within teams, and for cultural change, in medical education and in the NHS.

13. Is the supplementary reporting complete (e.g. trial registration; funding details; CONSORT, STROBE or PRISMA checklist)?

Yes – SRQR checklist has been included, but it is worthwhile including in the text that this has been followed also (in the Methods section, with reference)

Author response: We have amended our paper to include a line in the Methods section referencing the SRQR checklist (page 7; paragraph 3). Thank you.

14. To the best of your knowledge is the paper free from concerns over publication ethics (e.g. plagiarism, redundant publication, undeclared conflicts of interest)?

Yes

15. Is the standard of written English acceptable for publication?

Yes

REVIEWER 2

Dr. Johanna Spiers, University of Birmingham Comments to the Author:

*** Please find the example table that accompanies this review in the attached file *** Thank you for giving me the opportunity to review this paper, which is clearly organised, well-written and about an important topic. I think that, with the following revisions, the paper will make an important contribution to the literature.

I hope the amendments I've suggested don't feel overwhelming. While there are quite a few of them, I believe them all to be relatively minor. Well done on a great paper.

Abstract: This is clearly written - the study objectives are laid out well and each section feels thorough and full. This is an important topic and a unique sample, having been captured at this difficult time in history. Well done.

Author Response: Thank you for the positive feedback.

Introduction: This is another well-written section. You make a clear argument for why this topic is important given the conditions med students and junior doctors were working in at this time. One point I would make is that there has been some high-quality qualitative work on junior doctors working during the pandemic:

Spiers, J., Buszewicz, M., Chew-Graham, C., Dunning, A., Taylor, A. K., Gopfert, A., ... & Riley, R. (2021). What challenges did junior doctors face while working during the COVID-19 pandemic? A qualitative study. *BMJ open*, 11(12), e056122.

As well as other qualitative papers looking at NHS workers experiences during the pandemic more generally - for example:

Newman, K. L., Jevé, Y., & Majumder, P. (2022). Experiences and emotional strain of NHS frontline workers during the peak of the COVID-19 pandemic. *International Journal of Social Psychiatry*, 68(4), 783-790.

I would suggest citing these papers in your introduction and explaining how your paper builds on what is already in the literature.

Author response: We thank the Reviewer for their positive comments. Please also see our response to Reviewer 1, comment 3 above. We have now included this work and these studies in our introduction and referenced them as well. We have also noted that our study adds to the current literature, as it is the only qualitative study, to our knowledge, that has examined the mental health of UK-based medical students and junior doctors alongside each other during COVID-19 (page 20, paragraph 3).

To our knowledge, no qualitative studies have simultaneously examined the mental health of medical students and junior doctors during the COVID-19 pandemic.

If, as your thematic map (Fig. 1) suggests, you had three clear, distinctly worded research questions, I suggest presenting these as bullet points in the final para of your intro, to make it crystal clear to the reader what objectives you wanted to meet and how these questions fit into your thematic map.

Author response: This section has now been reformatted to fit with the reviewer's suggestion, as shown in the extract below (page 6; paragraph 2).

Specifically, this study addressed the following three research questions.

RQ1: *What are the key sources of occupational stress amongst medical students and junior doctors working during the COVID-19 pandemic?*

RQ2: *What barriers exist to accessing mental health support within this workforce?*

RQ3: *Have medical students and junior doctors received sufficient support to manage their work stress and mental health challenges during the COVID-19 pandemic?*

Methods: Your inclusion criteria are a little confusing. Was it your intention to include med students from the outset, or was this a later decision? Can you reword the first two sentences of your method section - and indeed the corresponding section of the abstract - to clarify this?

Author response: We apologise if this was not clear to the reader. Our intention was to include medical students and junior doctors from the outset. We have revised our methods section to clarify the inclusion criteria within the methods section and amended the abstract. The inclusion criteria are as follows (page 6, paragraph 3):

The inclusion criteria for the study were medical students and junior doctors working in the UK, during the summer of 2021. For the present study, 'junior doctor' refers to a doctor who graduated from medical school in the last five years, and who's currently enrolled on a medical training programme. Medical students were only invited to take part if they were in 4th year onwards. This cut-off was selected as these students have experience of clinical placement, and they faced significant disruption to their training, in close proximity to qualification.

Did you not carry out face-to-face interviews because of the pandemic? If so, please make that clear.

Author response: Thank you for the comments. We have added in a line to clarify that virtual interviews were conducted due to restrictions on social contact. We have inserted this revised section below, for convenience (found on page 7, paragraph 2).

Interviews took place over Zoom™, to adhere to the social distancing guidelines in place (July-August 2021) ⁴⁵.

The interviews seem rather short for a qualitative study on such an emotive topic. Can you comment on why this was?

Author response: Thank you for this comment. We were concerned about burdening participants with an extensive interview in amongst their work pressures during the COVID-19 pandemic, and between their otherwise busy schedules. Within our participant information sheet, we stated that the interviews would last approximately 40 minutes. Importantly, this 36-minute average does not include the introductions and rapport-building before the start of the main interview, or the debrief afterwards. Elsewhere, similar qualitative studies with healthcare professionals have reported

interview times of between 22 and 55 minutes (e.g. Bianchi et al. 2016; <https://bmjopen.bmj.com/content/6/9/e012598>), and 'up to 40 minutes' (e.g. Kuek et al. 2022; <https://www.ncbi.nlm.nih.gov/pmc/articles/PMC9341377/>).

The Braun and Clarke paper you cite is quite old now. B&C have published many more recent papers in which they develop and further explicate thematic analysis. Did you consult any of those papers? I would particularly suggest reading this paper:

Braun, V., & Clarke, V. (2019). Reflecting on reflexive thematic analysis. *Qualitative research in sport, exercise and health*, 11(4), 589-597.

Author response: Thank you for your recommendation. We have now reviewed the most recent Braun and Clarke paper, and updated our citation to reflect this.

Having read your results section, I feel that your themes are perhaps at the 'domain' level rather than the 'theme' level (as described in the above paper) as they are about a shared topic rather than shared meanings - they capture a diversity of meaning within related areas. Can you comment on this?

Author response: Thank you for this comment. Based on comment 6 of Reviewer 1 and Reviewer's 2 comment here, we appreciate that the presentation of our thematic map was not clear and we think inserting our research questions within the map seems to have caused confusion. We have now only presented our themes and the subthemes in a simpler and clearer way (please see revised thematic map). As stated in our response to comment 6 of Reviewer 1, from our thematic analysis, there were five themes -i.e. patterns of shared meaning- that emerged. These were central concepts that illustrated patterns across the data and across the questions that were asked of the participants, rather than 'domains' (e.g. a summary of participants responses to a particular question). As stated the research questions were guides to develop the interview schedule and were not questions asked directly to the participants. We have now revised our thematic map- and removed the presentation of our research questions which were there for illustrative purposes as to how our themes 'mapped' onto the research questions of the study.

I see that you mention data saturation, a term which comes from grounded theory and refers to the conclusion of a process in which researchers analyse data simultaneously to interviewing, and use their early theories and concepts to theoretically sample subsequent participants and interviews so that those theories and concepts can be tested and built upon. Achieving data saturation in a thematic analysis with only 15 participants seems quite ambitious. Can you say more to explain how you define saturation and how you determined that you had achieved it?

Author response: Thank you for this comment. Whilst the term data saturation originated in grounded theory it has emerged in several other qualitative methodologies. Within this research, we posited that data saturation was achieved as no new themes arose from the final 3 interviews. Despite this, we agree with the researcher, that it may be 'optimistic' to definitely suggest that data saturation was achieved when discussing complex phenomena with 15 interviews, and that more depth may have been acquired with further interviews. However, other research has found that it is challenging to say what sample size is sufficient to reach theoretical saturation and that many factors, including homogeneity of the sample, scope, data collection methods etc., can all impact on this. In the study of Guest et al. 2005, the authors concluded that most developed codes (88%) in their analysis of 60 qualitative interviews were established by the time 12 interviews had been completed (<https://journals.sagepub.com/doi/10.1177/1525822X05279903>).

We have, however, reworded our paper to more accurately summarise our sampling method. This section now reads (page 8; paragraph 1):

Data analysis took place in parallel with interviews, which meant that the researcher could closely follow emerging themes and confirm when no new themes were arising⁴⁸.

Results: Table 1 feels rather quantitative to me, and is a little hard to read. I'm not sure the percentages are meaningful in a qualitative study with a small sample. More typically, tables of participant demographics in qualitative studies are presented with participant ID numbers in the left-hand column and then further columns for other details - in this case, training level, place of work, gender ID and ethnicity. I've attached an example - unless you feel this compromises the anonymity of your participants, I would suggest presenting the table like this. If this would compromise confidentiality, I would still suggest reformatting this table in some way to make it less number-focused and more readable.

Author response: Thank you for this comment and for the table that you attached. The participant demographics table has now been redesigned to the reviewers suggested format, and a section is inserted below for reviewing (please see revised Table 1; page 9).

Participant ID	Training level	Place of work	Gender Identity	Ethnicity
#1	5 th year medical student	North-West England	Male	White-British
#2	5 th year medical student	North-West England	Male	White-British
#3	4 th year medical student	North-East England	Male	White-British
#4	FY1	Scotland	Male	White-British
#5	FY1	Midlands England	Male	White-British

I also feel that your thematic map needs a little further explanation. Where did these research questions come from? Are these the RQs that informed your study, or questions you asked participants in your interviews? The third question (have med students and JDs received an adequate level of support to manage their work-stress and mental health challenges during the pandemic) doesn't seem to match up to the third objective you currently list in your intro (how educational and health systems have adapted their systems to support students and JDs). The question in the thematic map and abstract focuses on the participants - the wording in the intro focuses on the systems. Can you clarify?

Author response: We have revised the thematic map to align more closely with the reviewer's suggestions. The research questions informed the study and the interview guide (Appendix 1), they were not specific questions we directly asked the participants. We have now updated our introduction to reflect this (page 6, paragraph 1).

We also acknowledge that there was an error in the submission, and that the focus of our 3rd RQ was on the participants' views and experiences of support during the pandemic. Many thanks for outlining this.

An explanation of what the various straight and dotted lines in the thematic map indicate would be useful too.

Author response: Thank you. The thematic map has been redesigned, based on the comments of Reviewers 1 & 2 above, to make clearer and easier to understand.

Theme 1: I would like to see the claim about high workloads supported with a quote from one of the junior doctors as well as the quote from the med student.

Author response: We have several quotes from junior doctors regarding their significant workloads that we could use to evidence this point, including the quote below. However, we are currently struggling with the word count so have decided not to include it in the main paper.

"(...) it definitely, definitely is a stressful, kind of demanding job and I can definitely appreciate that it probably would, compared to a general population or a normal job, just because you're around sick people, and most of the time you're well supported but it's when you have these kind of out of hour, where there's slightly less people and if all things go wrong and there's multiple sick people you might not necessarily have the ideal support that you would normally, and then that definitely would have its knock on effects yeah, and I think you often... especially for those first 4 months you're still thinking about stuff, and should I have done that differently or... which is good for kind of learning, but it also means that you don't kind of switch off fully and you end up just thinking about work, which can be mentally draining at times" (#5, FY1).

I suggest keeping quotes next to the comments about those quotes, as this helps the reader to follow your train of thought. So I would place the quote from #2 after the first line of para 3 on page 9, and then move onto the comment and quote about resilience. This is an issue at other points in the results section, so watch out for other places where you could clarify which comments 'belongs' to which quote.

Author response: Thank you for this comment, we have now amended the results section, so the quotes are next to the accompanying text. However, while we tried to cut down the paper as much as possible without losing the richness of the results, we had to remove the quotes from in the text for theme 5 and add these into a Box figure. These quotes amount to around 200 words over the journal word limit. If the Journal allows, we would much prefer to have these included within the text as suggested.

It would also be good to see a quote to support the claim that personality traits of medics could add to occupational stress.

Author response: Thank you for the comment, we have amended the paper to include a quote to support this claim, attached below (page 11; paragraph 1).

"The people that medicine attracts can be perfectionists where they might be more vulnerable to negative experiences and working in medicine comes with its fair share of those" (#7, 4th year student).

I wonder if you can unpack the quote about resilience a little further. Note that the participant says medics are resilient 'because we have to be' and that resilience 'just happens to you'. This data is ambiguous, perhaps suggesting a culture in which it is unacceptable NOT to claim resilience, rather than one in which genuine resilience is nurtured. What do you think?

Author response: Thank you for this comment. We have now redrafted this section, to be more explicit in describing the work cultures around resilience (please see page 11, paragraph 2). This section has been attached below.

Participants explained that you have to be resilient to practise medicine, to cope with the workload and the emotional aspects of the job. In some cases, participants alluded to a socialisation process which begins upon entry to medical school, where students must claim or develop the resilience that the profession requires.

Theme 2: It would be good to see a quote which evidences your point about the missed education being especially hard for the JDs who graduated early - at present, I think you have two quotes from

participants who felt they wouldn't have been ready no matter what? If I'm wrong, please clarify which quote comes from one of the early graduates and provide more context.

Author response: We feel that we have included a quote from an FY1 to evidence this. This participant stated that they will feel less prepared when working in ED in the future due to disruption to their undergraduate degree, which meant that they never received their placement in emergency medicine (page 12; paragraph 1).

Concerns around preparedness were described, specifically where aspects of education had been impaired or missed altogether. This point is particularly salient for the sub-cohort of junior doctors who graduated early to offer their services during the pandemic.

"I've still never worked in the Emergency Department (ED), I've never even been there as a medical student, so when it eventually comes to me being in ED, I'll definitely feel less prepared in that regard" (#5, FY1).

Theme 3: This is a well-organised and well-evidenced theme, well done.

Author response: Thank you for the positive feedback.

Theme 4: While this is another well-organised and well-written theme, it is striking to me that there are so many more positive quotes than quotes about the challenges of being supported and coping during COVID-19 (especially given your claim in the discussion that most participants felt there was less support during the pandemic). Was it the case that participants were more positive than negative about support and coping during this time? (Fantastic, if so!) Or does this balance need redressing a little?

Author response: Thank you for this comment. The majority of participants suffered from a loss of support during the pandemic due to restrictions to social contact. We have now included a quote to address this imbalance and updated the pretext introducing this quote to further outline this. Please see below and on page 16, paragraph 3 of the manuscript.

Simultaneously, participants discussed how their usual methods for managing stress had been hindered by the COVID-19 pandemic. Many participants described feeling isolated and went extended periods without seeing their close friends and family, whilst several FY1s found it difficult to integrate into new work settings.

"When lockdown was happening, I couldn't leave, and I think it got to the point where I was getting quite stressed about if my parents get sick, or my grandparents got sick, I wouldn't be there" (#15, FY3).

Theme 5: The first quote in this theme only seems to evidence your initial point, i.e. that extra efforts were made to support staff. Can you move this quote up so it appears underneath that comment and then also include a quote to illustrate your second very important point about the need for further support?

Author response: Thank you, we have now included an additional quote to outline this second point. We have inserted this quote below, for convenience.

"I think X university do it, they have mental health days where you can sort of, no questions asked, 'I'm not going in today', and you can just email them and say that, and you have a certain number throughout the year" (#8, 5th year student). (Please see page 18, paragraph 2)

The data in this theme is really strong and presents a worrying picture of mental health support in the NHS. Taken in conjunction with the quote in theme 1 where a participant says that doctors 'have to'

be resilient, this is further evidence for the toxic culture that my colleagues and I have reported in various papers:

Riley, R., Spiers, J., Buszewicz, M., Taylor, A. K., Thornton, G., & Chew-Graham, C.

(2018). What are the sources of stress and distress for general practitioners working in England? A qualitative study. *BMJ open*, 8(1), e017361.

Riley, R., Buszewicz, M., Kokab, F., Teoh, K., Gopfert, A., Taylor, A. K., ... & Chew-Graham, C. (2021). Sources of work-related psychological distress experienced by UK-wide foundation and junior doctors: a qualitative study. *BMJ open*, 11(6), e043521.

Author response: Thank you for providing us with these references, they were highly relevant to this paper and have been cited frequently. Specifically, within our discussion (page 19, paragraph 2), we have referred to damaging social norms within medicine that encourage presenteeism and sustain medics' burnout and poor mental health.

Discussion: You've said in the discussion that 'a majority of participants indicated that the pandemic had limited their access to support networks and coping strategies'. However, the findings suggest that there was more support available, plus most quotes in theme 4 suggest that participants were able to find ways to cope and be supported. This currently feels like a bit of a mismatch. Can you clarify?

Author response: Thank you, we have discussed and addressed this point. Please see our response to the comment of Reviewer 2 above, regarding Theme 4.

I don't agree with your second limitation, ie that the participants would find it hard to contextualise stress outside of the pandemic - your study is about medical students' and JDs' stress during the pandemic. I would suggest that this subject matter precludes any contextualisation away from COVID-19. I recommend deleting this limitation as I think you're doing yourself a disservice here.

Author response: We thank the reviewer for this comment, on reflection we agree and have removed this limitation.

While the protective measures you've suggested all sound good in theory, I wonder if you can comment on how realistic they are in practice? This paper might be useful to you here:

Riley, R., Kokab, F., Buszewicz, M., Gopfert, A., Van Hove, M., Taylor, A. K., ... & Chew-Graham, C. (2021). Protective factors and sources of support in the workplace as experienced by UK foundation and junior doctors: a qualitative study. *BMJ open*, 11(6), e045588.

Author response: Thank-you for the comment and the recommended paper. We have made several updates to our discussion to introduce possible interventions which are not resource intensive. Within these updates, we have cited Riley's paper (2021) which suggests that health systems look to build on pre-existing support networks within organisations (section attached below and page 21; paragraph 2 of the manuscript). Considering whether and how realistic these recommendations would require a more in-depth analysis, alongside an economic evaluation, which are beyond the scope of this study. However, the examples of good practice and policies and practices put in place in workplaces, demonstrate that these programmes and interventions are feasible.

Consequently, interventions which foster peer support must be considered, and existing support from colleagues, supervisors and teams should be utilised²². Elsewhere, good practices have included 'buddying-up' schemes to reduce isolation and peer support groups where medics can discuss the emotional impact of their work^{56 65}.

A few proofreading points:

This paper, while mostly well-written, contains some awkward-sounding sentences and grammar/punctuation errors. I would suggest running the paper through the free online version of Grammarly to iron those issues out - but here are some issues I've spotted.

Author response: Thank you for your suggestion. We have run the paper through Grammarly as suggested and also proofread again to spot any typos and grammatical errors.

You've got a stray comma after 'junior doctors' in the first line of the second bullet point of the article summary. I would also suggest breaking this second bullet point into two sentences: '...training opportunities. It provides...' There are several other misplaced or missing commas in this paper, so do look out for those.

Author response: We have run the paper through Grammarly as suggested and also proofread again to spot any typos and grammatical errors.

This sentence: 'Prior to arranging interviews, participants were made aware of the requirements and purposes of the research, as well as several ethical assurances, so that they could make an informed decision on participation' reads a little strangely. I would suggest '...of the research, and were given assurances about the ethics of the study so that they could make an informed decision about participation.'

Author response: This wording within the ethics section has been redrafted to address a previous comment. Please see our response to comment 5 of Reviewer 1 above.

REVIEWER 3

Dr. Randall Waechter, St George's University Comments to the Author:

This is an important topic of exploration given the pre-existing high rates of stress, burnout, and related mental illnesses among medical students and junior physicians (as well as physicians as a whole, compared to the non-physician population) prior to the COVID-19 pandemic. Medical educators and healthcare administrators need to have in-depth knowledge of how the pandemic impacted medical students and physicians so that effective support systems can be established for future health emergencies, whether at the local, regional, national, or international level. This manuscript provides an important contribution to the base of knowledge and I commend the authors for pursuing it. It was particularly enlightening to see the responses from the participants regarding stigmatization around mental illness, which confirms the amount of education and awareness work yet to be done within the healthcare field, let alone among the general population.

Author response: We thank the Reviewer for your time taken to review the paper, and to offer your expert insights.

There are a few limitations to the study, which must be taken into account when interpreting and applying the results.

1. The study aims was to "to develop a greater understanding of the work-stressors experienced by medical students and junior doctors during the pandemic....and assess how educational and health services have adapted their systems to support medical students and junior doctors during the COVID-19 pandemic". I suggest the authors add "in the UK" to clarify for the reader that this study applies specifically to that country.

Author response: Thank you for the feedback, the study aims have now been amended to include “in the UK”.

2. As per the above, I also suggest the authors add: "in the UK" to the title of the manuscript.

Author response: Similarly, the title has now been amended to include “in the UK”. Thank you.

3. The sample size is relatively small, and the sampling technique is not randomized. The authors state that selection response bias cannot be ruled out, but perhaps a more explicit statement about the sample not necessarily representing the experience of physicians across the UK is warranted.

Author response: We thank you for this comment. We have amended this section so that it now reads (page X; paragraph Y):

Additionally, whilst snowball sampling proved a useful recruitment technique, selection and response biases cannot be excluded ⁶², and the sample may not represent the experience of medics across the UK.

4. The authors provide suggestions for increasing access to occupational health care for medical students and junior doctors. There is good evidence that changes to the way medical training is provided is also helpful in reducing burnout and stress among students (e.g., Bloodgood RA, Short JG, Jackson JM, Martindale JR. A change to pass/fail grading in the first two years at one medical school results in improved psychological well-being. *Academic medicine : journal of the Association of American Medical Colleges.* 2009;84(5):655-62; Reed DA, Shanafelt TD, Satele DW, Power DV, Eacker A, Harper W, et al. Relationship of pass/fail grading and curriculum structure with well-being among preclinical medical students: a multi-institutional study. *Academic medicine : journal of the Association of American Medical Colleges.* 2011;86(11):1367-73; Rohe DE, Barrier PA, Clark MM, Cook DA, Vickers KS, Decker PA. The benefits of pass-fail grading on stress, mood, and group cohesion in medical students. *Mayo Clinic proceedings.* 2006;81(11):1443-8). There is also evidence that behavioral programs instituted in medical schools can provide students with the skills to support resilience and the authors may want to mention this (e.g., Yiu V. Supporting the well-being of medical students. *CMAJ: Canadian Medical Association Journal.* 2005;172(7):889-890).

Author response: Thank-you for these comments. We appreciate there are a number of different actions that can be taken .We have included the suggestion of a behavioural programme to support medical students’ resilience, which is pertinent to medical students training and general but much needed during the pandemic as well. We have added Yiu’s reference to our paper when discussing training interventions (page 21; paragraph 3).

“Educational and training interventions may be adopted, which can encourage help seeking behaviours and boost medics’ resilience ^{64 66 67}”.

Reviewer: 1

Competing interests of Reviewer: I declare I have no competing interests

Reviewer: 2

Competing interests of Reviewer: N/A

Reviewer: 3

Competing interests of Reviewer: I declare no competing interests in reviewing the manuscript titled:

“Medical students and junior doctors’ perceptions and experiences of occupational stress and mental health during the COVID-19 pandemic” bmjopen-2022-065639.

VERSION 2 – REVIEW

REVIEWER	Creese, Jennifer University of Leicester, Department of Health Sciences
REVIEW RETURNED	26-Oct-2022

GENERAL COMMENTS	Commendations to the authors for engaging so thoroughly with all three sets of reviewer comments. I find that the paper has been greatly strengthened, and I believe it makes a carefully-crafted and persuasive contribution both to literature on doctor and medical student wellbeing in general, to the historical record on COVID-19 experiences, and to future pandemic-proofing within the health system. I am happy to recommend proceeding to acceptance.
--

REVIEWER	Spiers, Johanna University of Birmingham, College of Medical and Dental Sciences
REVIEW RETURNED	27-Oct-2022

GENERAL COMMENTS	Thank you for the chance to re-read this important paper, which has been much improved. Well done to the authors. However, I'm afraid I do have a three more small points which I think need to be addressed before the paper can be accepted for publication.  1. On page 5, para 3, you say that there has been more quantitative than qualitative work on healthcare workers' mental health during the pandemic. However, you then cite four quant papers and seven qualitative papers, invalidating your wording. Please amend to say that both qual and quant researchers have explored this topic and then state what your paper adds to the existing picture. 2. The authors have placed the quotes from theme five in a box due to concerns about word count. However, in my experience, journal editors are less concerned about word count during the revision stage. It feels quite inconsistent to have the rest of the quotes inserted within the text and these ones in a box, so I hope the BMJ Open editors will be in agreement with you, as the authors, and me, as the reviewer, that placing the quotes back into the main text is a better option. 3. Note that there is a space between 'support' and a full stop on line 2 of page 18. Plus the sentence after this is poorly worded as it's not clear what the word 'these' is referring to. I think editing these sentences so that they read as follows would be better, and would save you some words: "However, some participants felt that they would have benefitted from receiving further support, such as more opportunities for rest and recovery, peer support and more accessible mental health services."
---

	Otherwise, I think this paper is ready for publication. Many thanks.
REVIEWER	Waechter, Randall St George's University
REVIEW RETURNED	06-Nov-2022

GENERAL COMMENTS	Then manuscript is much-improved and it was a pleasure to read it again. Please clarify what this sentence means: "Initially, purposeful sampling of participants was implemented..." Please explain what criteria were used to stop the snowball sampling at 15 participants. Given the acknowledged limitation that most of the participants were white males, why not recruit more participants with a more diverse background to address this limitation? If further recruitment is not undertaken, I think it is worthwhile to spend more time discussing the limitation of lack of diversity in the study sample. Is it reasonable to suspect that medical students and physicians of different gender and racial background would have experienced different and/or additional stressors during the COBID-19 pandemic? The addition of the research questions adds structure and aims to the manuscript. However, the authors do not mention them again after presenting them into the introduction. How were they addressed by the results of the study? This should be a major component of the discussion section and perhaps even the results section should be structured around these questions.
---

VERSION 2 – AUTHOR RESPONSE

Reviewer: 1

Dr. Jennifer Creese, University of Leicester

Comments to the Author: Commendations to the authors for engaging so thoroughly with all three sets of reviewer comments. I find that the paper has been greatly strengthened, and I believe it makes a carefully-crafted and persuasive contribution both to literature on doctor and medical student wellbeing in general, to the historical record on COVID-19 experiences, and to future pandemic-proofing within the health system. I am happy to recommend proceeding to acceptance.

Author's Response: Thank you very much for your time reading and reviewing our paper. Your comments have been so helpful in improving our manuscript.

Reviewer: 2

Dr. Johanna Spiers, University of Birmingham

Comments to the Author: Thank you for the chance to re-read this important paper, which has been much improved. Well done to the authors. However, I'm afraid I do have a three more small points

which I think need to be addressed before the paper can be accepted for publication.

1. On page 5, para 3, you say that there has been more quantitative than qualitative work on healthcare workers' mental health during the pandemic. However, you then cite four quant papers and seven qualitative papers, invalidating your wording. Please amend to say that both qual and quant researchers have explored this topic and then state what your paper adds to the existing picture.

Author's Response: Thank-you for outlining this. We have now amended this section to address this, the revised paragraph reads:

The mental health burden of junior doctors and medical students working during the pandemic has been the subject of interest in quantitative (1-5) and qualitative research studies(5-12). However, to the author's knowledge, no qualitative studies have simultaneously examined the mental health of medical students and junior doctors during COVID-19. As such, this paper can offer valuable insights into these groups' relative stress, mental wellbeing and access to support during this difficult transitional period from student to doctor (13), and in unprecedented circumstances. This qualitative study captures the voices of these groups at a distinct and testing time, amongst rising rates of COVID-19 in the summer of 2021 and the long-term impact of COVID-19 on healthcare workers' wellbeing remains unclear (14, 15). Consequently, continued efforts to protect medical students' and junior doctors' wellbeing are warranted. In doing so, retention issues may be addressed, which threaten patient safety, and the future of health services (16).

2. The authors have placed the quotes from theme five in a box due to concerns about word count. However, in my experience, journal editors are less concerned about word count during the revision stage. It feels quite inconsistent to have the rest of the quotes inserted within the text and these ones in a box, so I hope the BMJ Open editors will be in agreement with you, as the authors, and me, as the reviewer, that placing the quotes back into the main text is a better option.

Author's Response: Thank-you for supporting us with this, the journal has acknowledged this suggestion and is happy for us to place the quotes within the text. We have amended the manuscript with the quotes of theme five embedded in the text.

3. Note that there is a space between 'support' and a full stop on line 2 of page 18. Plus the sentence after this is poorly worded as it's not clear what the word 'these' is referring to. I think editing these sentences so that they read as follows would be better, and would save you some words:

"However, some participants felt that they would have benefitted from receiving further support, such as more opportunities for rest and recovery, peer support and more accessible mental health services."

Otherwise, I think this paper is ready for publication. Many thanks.

Author's Response: Thank you for this comment and for your suggested revision, which we are happy to accept. We have now replaced this section with the proposed wording above. Also, thank you so much for the time taken to help us to improve our paper, it is greatly appreciated.

Reviewer: 3

Dr. Randall Waechter, St George's University

Comments to the Author:

Then manuscript is much improved and it was a pleasure to read it again.

Author's Response: We thank you for your time to review our paper and for your very constructive comments, which have helped improve our paper.

Please clarify what this sentence means: "Initially, purposeful sampling of participants was implemented..."

Author's Response: Thank you for this comment. We have amended this section, to be clearer. It now reads:

"Initially, purposeful sampling of participants was implemented, where the lead researcher reached out to personal contacts and acquaintances who may have been available and willing to participate. Following this, participants were recruited using snowball sampling (17, 18)."

Please explain what criteria were used to stop the snowball sampling at 15 participants.

Author's Response: Our original plan was to interview approximately 15-20 participants (as stated in our ethical application; reference: PGT/SPS/2021/035/GLOB). We stopped recruiting after we had conducted 15 interviews with participants, because no new themes arose from the final 3 interviews. We were able to monitor this, as data analysis took place in parallel with interviews, which meant that the researcher could closely follow emerging themes and confirm when no new themes were arising (please see our methods section). Previous research has found that it is challenging to say what sample size is sufficient to reach theoretical saturation and that many factors, including homogeneity of the sample, scope, data collection methods etc., can all impact on this. In the study of Guest et al. 2005, the authors concluded that most developed codes (88%) in their analysis of 60 qualitative interviews were established by the time 12 interviews had been completed (19). To clarify, we have added in our methods section what our initial aim was and why we stopped after conducting 15 participant interviews. The revised sections read:

"We aimed to conduct approximately 15-20 interviews". (please see Participant Recruitment section in the Methods)

“Data analysis took place in parallel with interviews, which meant that the researcher could closely follow emerging themes and confirm when no new themes were arising (20).” (please see Data Analysis Methodology section in the Methods)

Given the acknowledged limitation that most of the participants were white males, why not recruit more participants with a more diverse background to address this limitation?

If further recruitment is not undertaken, I think it is worthwhile to spend more time discussing the limitation of lack of diversity in the study sample. Is it reasonable to suspect that medical students and physicians of different gender and racial background would have experienced different and/or additional stressors during the COVID-19 pandemic?

Author’s Response: Further recruitment at this stage is not possible. This qualitative study captures the voices of these groups at a very distinct and testing time, amongst rising rates of COVID-19 in the summer of 2021. Recruiting new participants at this time-point, post-pandemic and with very different conditions in healthcare, socio-political landscape and in the educational sector would not provide comparable data. Furthermore, we feel that this point has been addressed by the following section.

Despite this, the representativeness and transferability of the findings may be limited, as most participants were males and of a white-British ethnicity. This is of particular concern as women, and ethnically diverse medics were reported to have suffered from higher rates of anxiety, depression and stress during the COVID-19 pandemic (21-23).

The addition of the research questions adds structure and aims to the manuscript. However, the authors do not mention them again after presenting them into the introduction. How were they addressed by the results of the study? This should be a major component of the discussion section and perhaps even the results section should be structured around these questions.

Author’s Response: The research questions were guides to develop the interview schedule and were not questions asked directly to the participants. From our thematic analysis, there were five themes i.e. patterns of shared meaning- that emerged. These were the central concepts that illustrated patterns across the data and across the questions that were asked of the participants. As per the suggestions of the previous two reviewers, we revised our thematic map- and removed the presentation of our research questions which were there for illustrative purposes as to how our themes ‘mapped’ onto the research questions of the study. As the themes that emerged spanned across the research questions (that were guides to develop the interview schedule) the other Reviewers found this confusing. We also agree the revised document is much clearer, and would suggest keeping as is.

To clarify this however, we have added a sentence in our methods sections which now reads:

“Semi-structured, one-on-one interviews were conducted online. The research questions were used as guides to develop the interview schedule (Appendix 1).”(please see 3rd paragraph of our Participant Recruitment section in the Methods)

Reviewer: 1

Competing interests of Reviewer: I declare I have no competing interests

Reviewer: 2

Competing interests of Reviewer: N/A

Reviewer: 3

Competing interests of Reviewer: None

REFERENCES

1. Pandey U, Corbett G, Mohan S, Reagu S, Kumar S, Farrell T, et al. Anxiety, Depression and Behavioural Changes in Junior Doctors and Medical Students Associated with the Coronavirus Pandemic: A Cross-Sectional Survey. *J Obstet Gynaecol India*. 2020;1-5.
2. Zis P, Artemiadis A, Bargiotas P, Nteveros A, Hadjigeorgiou GM. Medical Studies during the COVID-19 Pandemic: The Impact of Digital Learning on Medical Students' Burnout and Mental Health. *Int J Environ Res Public Health*. 2021;18(1).
3. Essadek A, Gressier F, Robin M, Shadili G, Bastien L, Peronnet JC, et al. Mental health of medical students during the COVID19: Impact of studies years. *J Affect Disord Rep*. 2022;8:100318.
4. O'Byrne L, Gavin B, Adamis D, Lim YX, McNicholas F. Levels of stress in medical students due to COVID-19. *J Med Ethics*. 2021.
5. Dunning A, Teoh K, Martin J, Spiers J, Buszewicz M, Chew-Graham C, et al. Relationship between working conditions and psychological distress experienced by junior doctors in the UK during the COVID-19 pandemic: a cross-sectional survey study. *BMJ Open*. 2022;12(8):e061331.
6. Spiers J, Buszewicz M, Chew-Graham C, Dunning A, Taylor AK, Gopfert A, et al. What challenges did junior doctors face while working during the COVID-19 pandemic? A qualitative study. *BMJ Open*. 2021;11(12):e056122.
7. Warren J, Plunkett E, Rudge J, Stamoulis C, Torlinski T, Tarrant C, et al. Trainee doctors' experiences of learning and well-being while working in intensive care during the COVID-19 pandemic: a qualitative study using appreciative inquiry. *BMJ Open*. 2021;11(5):e049437.
8. Byrne JP, Creese J, Matthews A, McDermott AM, Costello RW, Humphries N. '...the way it was staffed during COVID is the way it should be staffed in real life...': a qualitative study of the impact of COVID-19 on the working conditions of junior hospital doctors. *BMJ Open*. 2021;11(8):e050358.
9. Olah B, Radi BM, Kosa K. Barriers to Seeking Mental Help and Interventions to Remove Them in Medical School during the COVID-19 Pandemic: Perspectives of Students. *Int J Environ Res Public Health*. 2022;19(13).

10. Kromydas T, Green M, Craig P, Katikireddi SV, Leyland AH, Niedzwiedz CL, et al. Comparing population-level mental health of UK workers before and during the COVID-19 pandemic: a longitudinal study using Understanding Society. *Journal of Epidemiology and Community Health*. 2022;76(6):527.
11. Mutambudzi M, Niedzwiedz C, Macdonald EB, Leyland A, Mair F, Anderson J, et al. Occupation and risk of severe COVID-19: prospective cohort study of 120 075 UK Biobank participants. *Occupational and Environmental Medicine*. 2020.
12. Griffin L, Riley R. Exploring the psychological impact of working during COVID-19 on medical and nursing students: a qualitative study. *BMJ Open*. 2022;12(6):e055804.
13. Wilkinson C, Finn G, Crampton P. Responsibility with a Safety Net: Exploring the Medical Student to Junior Doctor Transition During COVID-19. *Medical Science Educator*. 2021;32(1):121-9.
14. Petrino R, Riesgo LG, Yilmaz B. Burnout in emergency medicine professionals after 2 years of the COVID-19 pandemic: a threat to the healthcare system? *Eur J Emerg Med*. 2022;29(4):279-84.
15. Zhou T, Xu C, Wang C, Sha S, Wang Z, Zhou Y, et al. Burnout and well-being of healthcare workers in the post-pandemic period of COVID-19: a perspective from the job demands-resources model. *BMC Health Serv Res*. 2022;22(1):284.
16. Lock FK, Carrieri D. Factors affecting the UK junior doctor workforce retention crisis: an integrative review. *BMJ Open*. 2022;12(3):e059397.
17. Noy C. Sampling Knowledge: The Hermeneutics of Snowball Sampling in Qualitative Research. *International Journal of Social Research Methodology*. 2008;11(4):327-44.
18. Naderifar M, Goli H, Ghaljaie F. Snowball Sampling: A Purposeful Method of Sampling in Qualitative Research. *Strides in Development of Medical Education*. 2017;14(3).
19. Guest G, Bunce A, Johnson L. How Many Interviews Are Enough?: An Experiment with Data Saturation and Variability. *Field Methods*. 2006;18(1):59-82.
20. Palinkas LA. Qualitative and mixed methods in mental health services and implementation research. *J Clin Child Adolesc Psychol*. 2014;43(6):851-61.
21. Morgan R, Tan HL, Oveisi N, Memmott C, Korzuchowski A, Hawkins K, et al. Women healthcare workers' experiences during COVID-19 and other crises: A scoping review. *Int J Nurs Stud Adv*. 2022;4:100066.
22. Regenold N, Vindrola-Padros C. Gender Matters: A Gender Analysis of Healthcare Workers' Experiences during the First COVID-19 Pandemic Peak in England. *Social Sciences*. 2021;10(2).
23. Qureshi I, Gogoi M, Wobi F, Chaloner J, Al-Oraibi A, Hassan O, et al. Healthcare Workers From Diverse Ethnicities and Their Perceptions of Risk and Experiences of Risk Management During the COVID-19 Pandemic: Qualitative Insights From the United Kingdom-REACH Study. *Frontiers in Medicine*. 2022;9.

VERSION 3 – REVIEW

REVIEWER	Spiers, Johanna University of Birmingham, College of Medical and Dental Sciences
REVIEW RETURNED	16-Nov-2022
GENERAL COMMENTS	Great work, I believe this paper is now ready to be published.